# Eating the brain - A multidisciplinary study provides new insights into the mechanisms underlying the cytopathogenicity of *Naegleria fowleri*

Ronald Malych[1], Filipe Folgosa[2], Jana Pilátová[3,4,5], Libor Mikeš[6], Vít Dohnálek[1], Jan Mach[1], Magdaléna Matějková[1], Vladimír Kopecký[3], Pavel Doležal[1], Robert Sutak[1]*

1 Department of Parasitology, Faculty of Science, Charles University, BIOCEV, Vestec, Czech Republic, 2 Instituto de Tecnologia Química e Biológica António Xavier, Universidade Nova de Lisboa, Oeiras, Portugal, 3 Faculty of Mathematics and Physics, Institute of Physics, Charles University, Praha, Czech Republic, 4 Lawrence Berkeley National Laboratory, Molecular foundry, Berkeley, California, United States of America, 5 Intitute of Parasitology, Biology Centre, Czech Academy of Science, České Budějovice, Czech Republic, 6 Department of Parasitology, Faculty of Science, Charles University, Prague, Czech Republic

* sutak@natur.cuni.cz

## Abstract

*Naegleria fowleri*, the causative agent of primary amoebic meningoencephalitis (PAM), requires increased research attention due to its high lethality and the potential for increased incidence as a result of global warming. The aim of this study was to investigate the interactions between *N. fowleri* and host cells in order to elucidate the mechanisms underlying the pathogenicity of this amoeba. A co-culture system comprising human fibrosarcoma cells was established to study both contact-dependent and contact-independent cytopathogenicity. Proteomic analyses of the amoebas exposed to human cell cultures or passaged through mouse brain were used to identify novel virulence factors. Our results indicate that actin dynamics, regulated by Arp2/3 and Src kinase, play a considerable role in ingestion of host cells by amoebae. We have identified three promising candidate virulence factors, namely lysozyme, cystatin and hemerythrin, which may be critical in facilitating *N. fowleri* evasion of host defenses, migration to the brain and induction of a lethal infection. Long-term co-culture secretome analysis revealed an increase in protease secretion, which enhances *N. fowleri* cytopathogenicity. Raman microspectroscopy revealed significant metabolic differences between axenic and brain-isolated amoebae, particularly in lipid storage and utilization. Taken together, our findings provide important new insights into the pathogenic mechanisms of *N. fowleri* and highlight potential targets for therapeutic intervention against PAM.

## Author summary

*Naegleria fowleri*, infamously known as the brain-eating amoeba, causes a rare but almost always fatal disease called primary amoebic meningoencephalitis (PAM). Infection

**Data availability statement:** All relevant data are within the manuscript and its Supporting Information files. The mass spectrometry proteomics data have been deposited to the ProteomeXchange Consortium via the PRIDE partner repository with the dataset identifier PXD056622. Custom Google colab script can be freely accessed at the following GitHub repository: https://github.com/vitdohnalek/Batch_AlphaFold2_multimer_v2.

**Funding:** The project was funded by CePaViP, provided by ERDF and MEYS CR (CZ.02.1.01/0.0/0.0/16_019/0000759 to RS) and by Charles University Grant Agency (171124 to RM). This work was also financially supported by the Portuguese Fundação para a Ciência e Tecnologia (FCT), PTDC/BIA-BQM/0562/2020 project to FF, MOSTMICRO-ITQB R&D Unit (references UIDB/04612/2020 and UIDP/04612/2020 to FF), and LS4FUTURE Associated Laboratory (LA/P/0087/2020 to FF), and by the Laboratory Directed Research and Development program at Lawrence Berkeley National Laboratory (LDRD 25-116 to JP). The funders had no role in study design, data collection and analysis, decision to publish, or preparation of the manuscript.

**Competing interests:** The authors have declared that no competing interests exist.

usually occurs during water activities in healthy people or during nose rinsing with contaminated water, when the amoeba migrates via the olfactory nerve to the brain. The incidence of PAM is expected to increase due to global climate change, which favors this thermophilic amoeba, as well as improved diagnostics in developing countries. Very little is known about how this free-living microorganism, naturally found in warm waters where it feeds on bacteria and other microbes, can become a deadly parasite when exposed to the host. Our multidisciplinary study, employing various biochemical and bioimaging techniques, has succeeded in showing how *N. fowleri* changes at the molecular (protein) level after contact with the host environment and has enabled the identification of potentially crucial virulence factors that can become targets for therapeutic intervention.

## Introduction

*Naegleria fowleri*, the causative agent of the rare primary amoebic meningoencephalitis (PAM), deserves research attention because of its extreme lethality and the high probability that its incidence will increase due to global warming [1,2]. Little is known about the mechanisms by which this amoeba, unlike its non-pathogenic relative *Naegleria gruberi*, survives host defenses, reaches the brain, and almost always causes death.

To date, several proteins have been identified that assist *N. fowleri* in the early stages of host invasion and during migration to the brain. One example is the *N. fowleri* glycosidase, which is involved in the degradation of intranasal mucus and helps *N. fowleri* to adhere to epithelial cells and further migrate to the brain [3]. Attachment to the nasal epithelium and direct contact with host cells is another part of amoeba cytopathogenesis. Glycoconjugates on the surface of *Naegleria* play an important role in host cell attachment, with their composition varying between non-pathogenic *N. gruberi*, *N. lovaniensis*, and pathogenic *N. fowleri* [4,5]. To minimize and potentially evade the host immune response during the invasion of the neuro-olfactory epithelium, *N. fowleri* disrupts tight junctions by degrading their proteins [6,7], adheres to the extracellular matrix [8] and breaks it down by secretion of proteases [9–11]. Once in the central nervous system, a number of different immune cells such as neutrophils and macrophages are recruited to prevent *N. fowleri* invasion. Neutrophils produce extracellular traps that immobilize *N. fowleri* and secrete myeloperoxidase, which generates reactive oxygen and nitrogen species that kill *Naegleria* [12,13]. *N. fowleri* evades its destruction by the activation of the antioxidant proteins such as glutathione peroxidase, peroxiredoxin and superoxide dismutase [13]. Macrophage amoebicidal activity differs among macrophage cell lines and the cytolysis of *N. fowleri* is dependent on macrophage activation [14,15].

Multiple studies have identified potential *N. fowleri* virulence factors such as proteases, pore-forming proteins, or actin [16–18], but none of these appear to be unique to *N. fowleri*. These also include recent analysis of extracellular vesicles produced by *N. fowleri* [19]. Importantly, the virulence of *N. fowleri* can be enhanced by long-term co-culture of amoebae with mammalian cells [20], but the exact mechanism of this phenomenon has not been described. Initial genomic and transcriptomic studies indicated potential virulence factors that would explain such behavior [21,22]. However, often the transcriptomic changes do not manifest in the cellular proteome, as it was clearly shown in the response to iron-dependent stress conditions showing completely different outcome at the proteomic and transcriptomic levels [23].

In this study, we aimed to deepen the understanding of the interactions between *N. fowleri* and host cells, the mechanisms of amoeba virulence, and yet unexplored virulence factors that may represent novel targets for chemotherapeutic intervention. After establishing a stable

co-culture system of *N. fowleri* with human cells that allows the quantification of amoeba cytopathogenicity, we investigated the mechanism of contact-dependent cytopathogenicity using inhibitors of actin and several protein kinases related to cellular processes such as motility and adhesion. We performed label-free proteomic analysis comparing axenic, bacteria co-cultured, mammalian cells co-cultured and brain-isolated *N. fowleri,* and identified several novel potential virulence factors. We investigated the differences in the secretomes and trogocytosis between axenic amoebae and long-term co-cultured amoebae with mammalian cells. Finally, utilizing Raman microspectroscopy we analyzed the cell composition of axenic *N. fowleri* and compared it to amoebae that were isolated from the mouse brain.

## Results

### Actin plays a significant role in *Naegleria fowleri* trogocytosis

To enable quantification of *N. fowleri* cytopathogenicity and visualization of ingestion of host cells by amoebae, we developed a co-culture system of amoebae labeled with non-toxic, cell-permeable fluorescent dye carboxyfluorescein succinimidyl ester (CFSE), which retains within the cells for long periods of time, together with tdTomato expressing human fibrosarcoma cells (HT1080). To quantify the cytopathogenic effect of *N. fowleri*, the percentage of CFSE-labeled amoebae containing tdTomato fluorescent protein from human cells was counted by flow cytometry (Fig 1A). The stability of the co-culture and the ingestion of fibrosarcoma cells by the amoebae were verified by live cell imaging and fluorescence microscopy (Fig 1B and S1 Fig and S1 Video).

Trogocytosis is one of the key cytopathogenicity mechanisms of *Naegleria* and requires dynamic remodeling of the actin cytoskeleton [24–26]. To verify the efficacy of our co-culture system, we analyzed the effect of actin inhibition on amoeba cytopathogenicity. We pretreated amoeba cells with CK-666 (Arp2/3 complex inhibitor), wortmannin (phosphoinositide 3-kinase inhibitor), piceatannol (tyrosine-protein kinase Syk inhibitor), and PP2 (tyrosine-protein kinase Src inhibitor) and incubated them with the fibrosarcoma cells for 3 hours. For *Naegleria* treated with CK-666 and PP2, we observed a decrease in the percentage of tdTomato-positive amoebae compared to the control cells and the cells treated with other inhibitors (Fig 1C, representative flow cytograms in S2 Fig). Previous research has demonstrated that inhibiting Arp2/3 complex impairs cell motility and phagocytosis in *N. gruberi* [27]. Furthermore, Src kinases are involved in the control of actin dynamics in numerous processes within mammalian cells [28,29], reinforcing the notion that actin remodeling may play a pivotal role in *N. fowleri* trogocytosis.

### Cellular proteomics of *Naegleria* co-cultures and a *Naegleria* brain isolate reveals novel candidate virulence factors

To obtain a comprehensive overview of the cellular response of *N. fowleri* to its prey, which could lead to the identification of novel virulence factors, we used label-free comparative proteomic analysis of *Naegleria* co-cultures with attenuated *Klebsiella aerogenes* or human fibrosarcoma cells HT1080 and amoebae isolated from the brains of infected mice. To study the interactions of the amoebae with bacteria, we selected short-term conditions (6 hours), while for the amoeba-fibrosarcoma co-culture, both short-term and long-term conditions (5 passages with host cells) were chosen. All conditions were compared to axenically cultured *Naegleria*. Complete proteomic datasets are available in S1 Table. We detected 3680 proteins across all conditions and further focused on only those proteins whose expression was more than 2-fold higher in the co-culture condition with statistical significance, or that were identified in all replicates of the co-culture condition and were not identified in any replicate of the

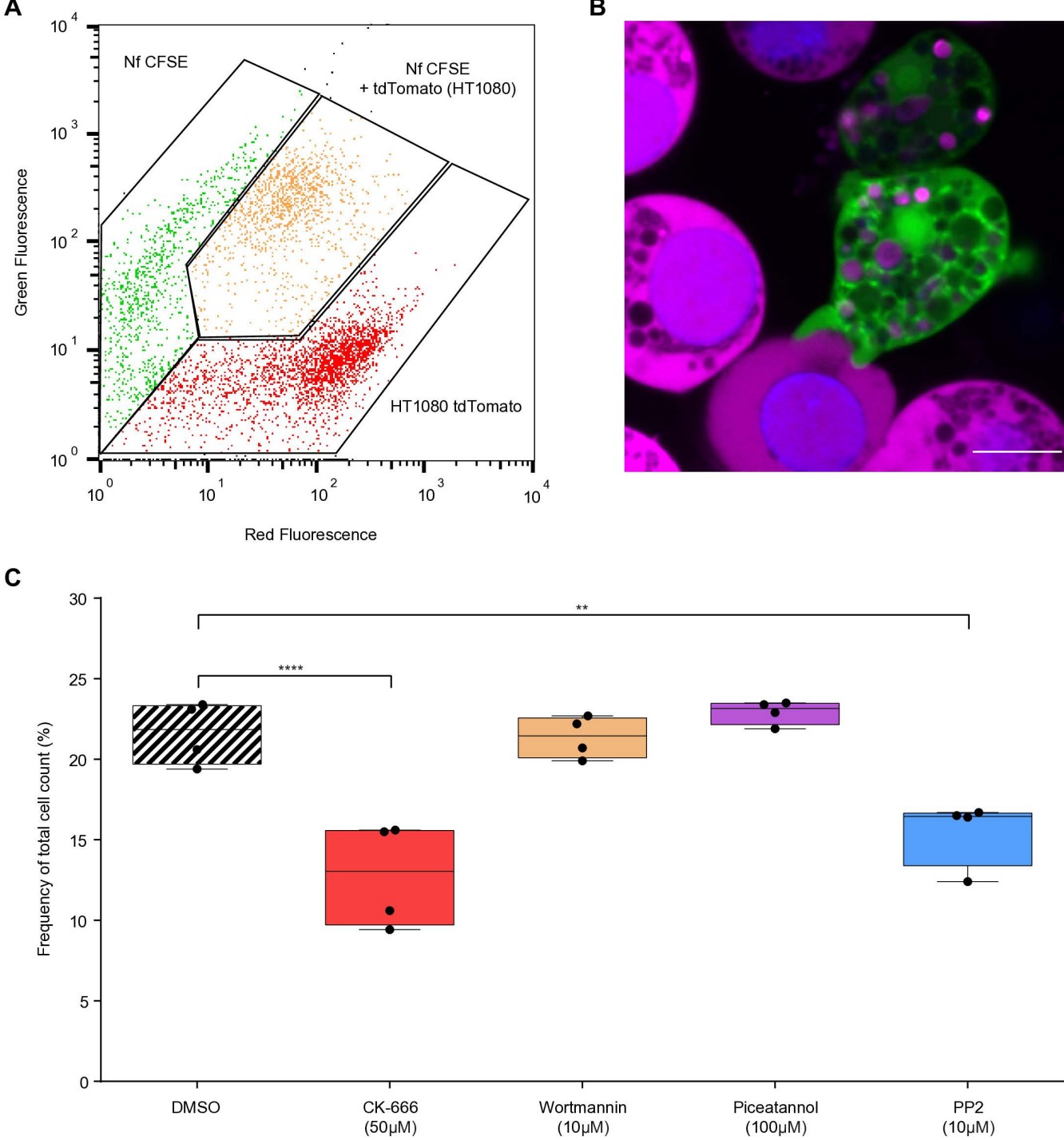

**Fig 1. *Naegleria fowleri* trogocytosis is affected by inhibition of actin and tyrosine-protein kinase Src.** (A) Representative flow cytogram of CFSE-labeled *Naegleria fowleri* (green) in co-culture with tdTomato expressing human fibrosarcoma cells HT1080 (red). The degree of trogocytosis was determined by the number of *Naegleria* with ingested cell parts, represented by the red fluorescence of the tdTomato (orange) after 3 hours of incubation. (B) Live imaging of CFSE-labeled *N. fowleri* (green) and HT1080 (magenta) co-cultures showing the process of *Naegleria* adhesion to human cells. Ingested parts of HT1080 cells are visible in *Naegleria* vacuoles. Nuclei were labeled with Hoechst 33342 (blue). Scale bar =10μm. (C) Box-plot graph representing effect of Arp2/3 complex inhibitor (CK-666) and different types of kinase inhibitors (wortmannin, piceatannol and PP2) on the percentage of CFSE-labeled *Naegleria* with ingested tdTomato from HT1080 cells. Data were collected from four individual experiments. Statistical significance was determined using an ordinary ANOVA and Dunnett's multiple comparison test. (** p-value<0.01, **** p-value<0.0001).

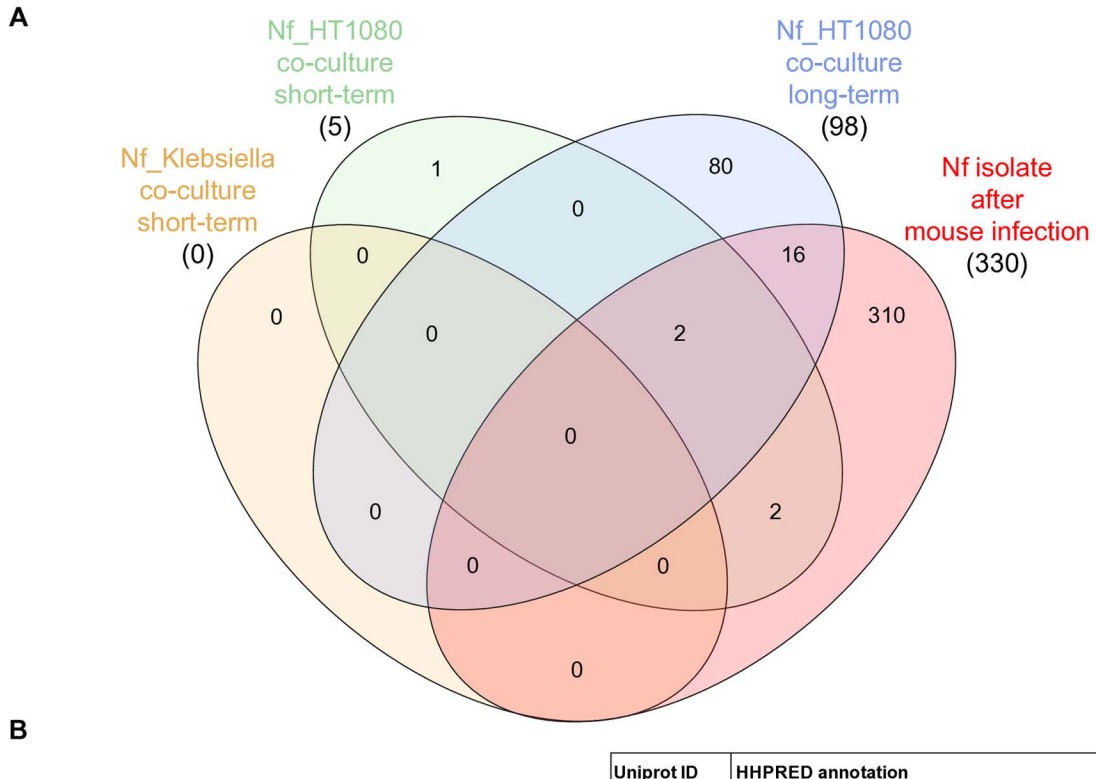

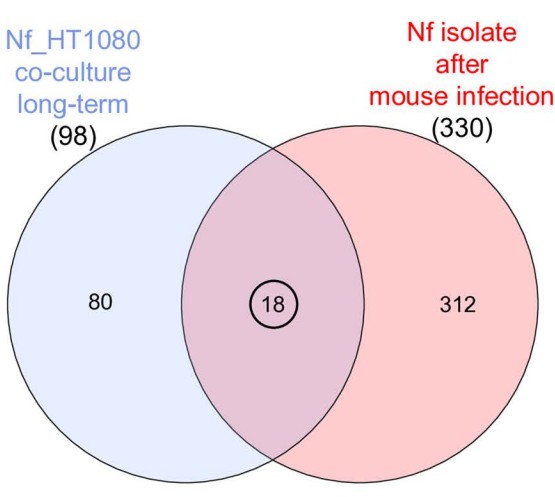

| Uniprot ID | HHPRED annotation |
|---|---|
| A0A6A5BWY3 | Angio-associated migratory cell protein |
| A0A6A5BL01 | Sterol carrier protein 2 |
| A0A6A5BTY6 | Coenzyme A-disulfide reductase |
| A0A6A5CFX1 | AAA family ATPase |
| A0A6A5CE35 | Rhodanese-related sulfurtransferase |
| A0A6A5CBW2 | Hypothetical protein |
| A0A6A5BGI7 | Sulfite oxidase |
| A0A6A5BG35 | DNA polymerase epsilon subunit D |
| A0A6A5BDE0 | Lysozyme |
| A0A6A5BD61 | S-adenosylmethionine-dependent methyltransferase |
| A0A6A5BFF9 | NPC1-like intracellular cholesterol transporter 1 |
| A0A6A5BDZ9 | Mitochondrial inner membrane protein OXA1L |
| A0A6A5BDQ5 | Unknown |
| A0A6A5C256 | U3 snoRNP-associated protein |
| A0A6A5C6T5 | Unknown |
| A0A6A5C679 | Cytosolic phospholipase A2 |
| A0A6A5C6G2 | Ubiquitin-protein ligase |
| A0A6A5C4B4 | Unknown |

**Fig 2. Proteomic changes in *Naegleria fowleri* co-cultured with bacterial or mammalian cells and amoebae isolated from infected mice.** (A) Venn diagram showing the overlap of upregulated proteins between short-term (6 h) *Naegleria* co-cultures with *Klebsiella aerogenes* or human fibrosarcoma cells HT1080, long-term (5 passages) co-culture with HT1080 cells and amoebae isolated from the brain of infected mice. (B) Venn diagram showing the overlap of 18 upregulated proteins between *Naegleria* long-term co-culture with human fibrosarcoma cells HT1080 and brain isolated amoebae. These proteins, listed in the table, include (marked in red) proteins involved in sulfur and lipid metabolism and lysozyme, which is discussed in the text as a potential novel virulence factor. Proteins were manually annotated by HHpred [70].

axenic culture. The Venn diagram in Fig 2A shows the number of upregulated proteins and their overlap between conditions. Both short-term co-cultures had a small number of upregulated proteins; no statistically significant proteins in the case of *Naegleria*-bacterial co-culture and five proteins in the case of *Naegleria*-fibrosarcoma co-culture, with only one upregulated protein unique to this condition (fatty acid desaturase A0A6A5C958 in fibrosarcoma co-culture). Long-term co-cultured and brain-isolated amoebae showed a more dramatic change in the proteome: 98 upregulated proteins in long-term *Naegleria*-fibrosarcoma co-culture and 330 in brain-isolated *Naegleria*. Two upregulated proteins were common to all conditions involving the mammalian host, A0A6A5BDQ5 and A0A6A5C4B4, both of unknown function. We further focused on the analysis of the 18 upregulated proteins that were shared between the long-term fibrosarcoma co-culture and brain-isolated amoebae as these are likely to be involved in the interaction with the host (all 18 proteins with HHpred annotation are listed in Fig 2B). From this dataset, we identified three proteins involved in the metabolism of sulfur or sulfur-containing compounds: coenzyme A disulfide reductase (A0A6A5BTY6), rhodanese-related sulfur transferase (A0A6A5CE35), and sulfite oxidase (A0A6A5BGI7). We also identified three proteins involved in lipid metabolism: sterol carrier protein 2 (A0A6A5BL01), NPC1-like intracellular cholesterol transporter 1 (A0A6A5BFF9), and cytosolic phospholipase A2 (A0A6A5C679).

One of the upregulated proteins from this dataset was predicted by HHpred to be lysozyme (A0A6A5BDE0), which was previously shown to be *N. fowleri*-specific [21]. The expression of the protein was 6.9-fold higher in long-term *Naegleria*-fibrosarcoma co-culture and 2.2-fold higher in brain-isolated amoebae. This lysozyme (here after named as Nf lysozyme) contains two domains: a lysozyme and a peptidoglycan binding domain (protein sequence and structure prediction is depicted in S3 Fig).

As mentioned above, 330 proteins are upregulated in the brain. Among the proteins with predicted function by HHpred (S2 Table and S4 Fig), 16 are cytoskeletal proteins related to migration in tissues, 11 are proteases/hydrolases probably important for virulence, and finally 11 are lipid metabolism proteins, indicating that the molecules are a source of energy for the amoebae in the brain, as we show in further results.

Among the proteins that were upregulated exclusively during the brain invasion and thus represent the most promising virulence factors, we identified cystatin (A0A6A5BXL1) as one of the proteins with the strongest responses in the brain-isolate condition (7.6-fold higher expression). Cystatins are a family of proteins that act as cysteine protease inhibitors. Two cystatins from *N. fowleri* were previously shown to be able to inhibit cysteine proteases and may play a role in defense against host immunity [30,31]. Cystatin identified in our proteomics study (here after named as Nf cystatin) consists of two active domains (protein sequence and structure predictions are shown in S5 Fig). To search for a potential target of inhibition, we used the AlphaFold2 multimer v.2 model to predict hypothetical interactions between Nf cystatin and cysteine proteases from *Naegleria fowleri* and human genomes. Out of 69 *N. fowleri* cysteine proteases, 7 had high predicted template modeling score (ipTM) values (> 0.8) with Nf cystatin (S3 Table). The top two hits with highest ipTM (0.87) were cathepsin L (A0A6A5BN68) and cathepsin F (A0A6A5C7W1; models shown in Fig 3A). Out of 237 *H. sapiens* cysteine proteases, 13 had a high interaction score with Nf cystatin (S4 Table), with the highest ipTM values (0.89) having cathepsin K (A0A7I2V4B1) and cathepsin L (P07711; models in Fig 3B).

Nfa1 (A0A6A5BCB0) is a protein proposed to be one of the major virulence factors of *N. fowleri* [32], and its expression has previously been demonstrated to be strongly affected by iron deficiency [23]. In our proteomic dataset, the expression of Nfa1 is not induced in the long-term fibrosarcoma co-culture and brain-isolated amoebae, nor are other suggested

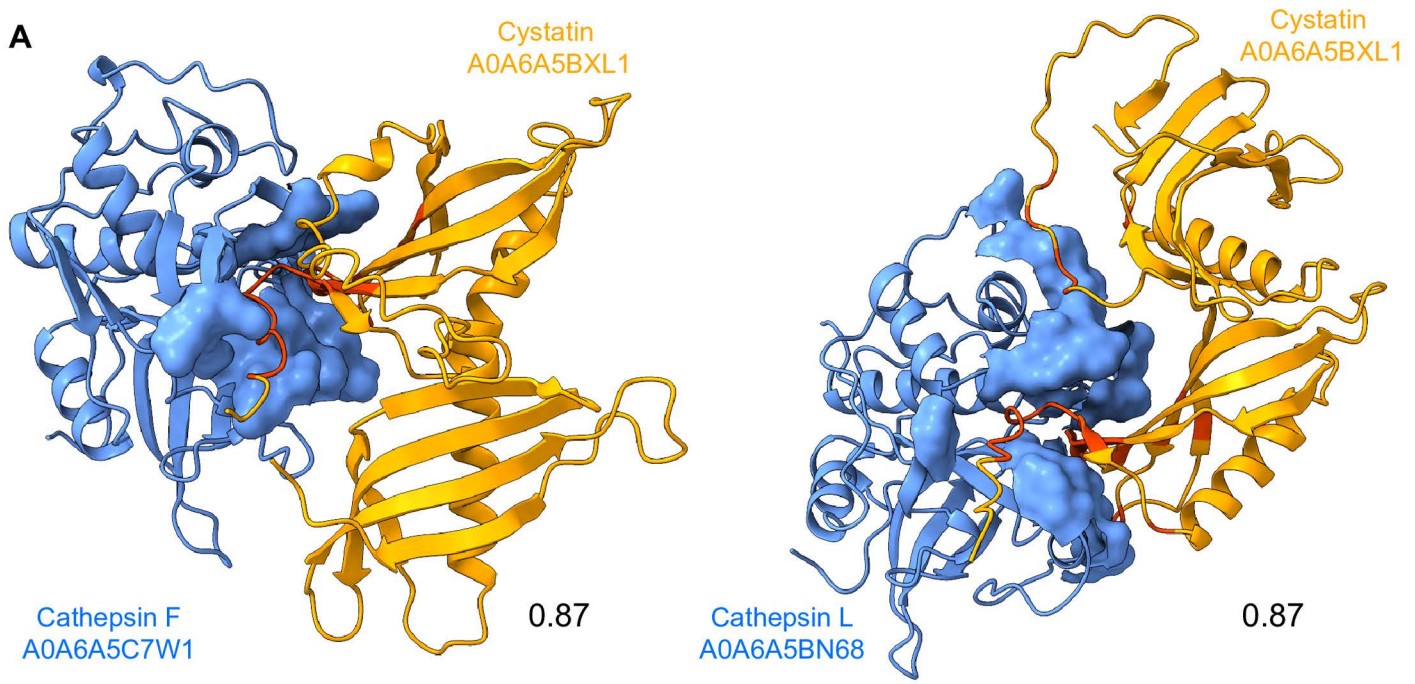

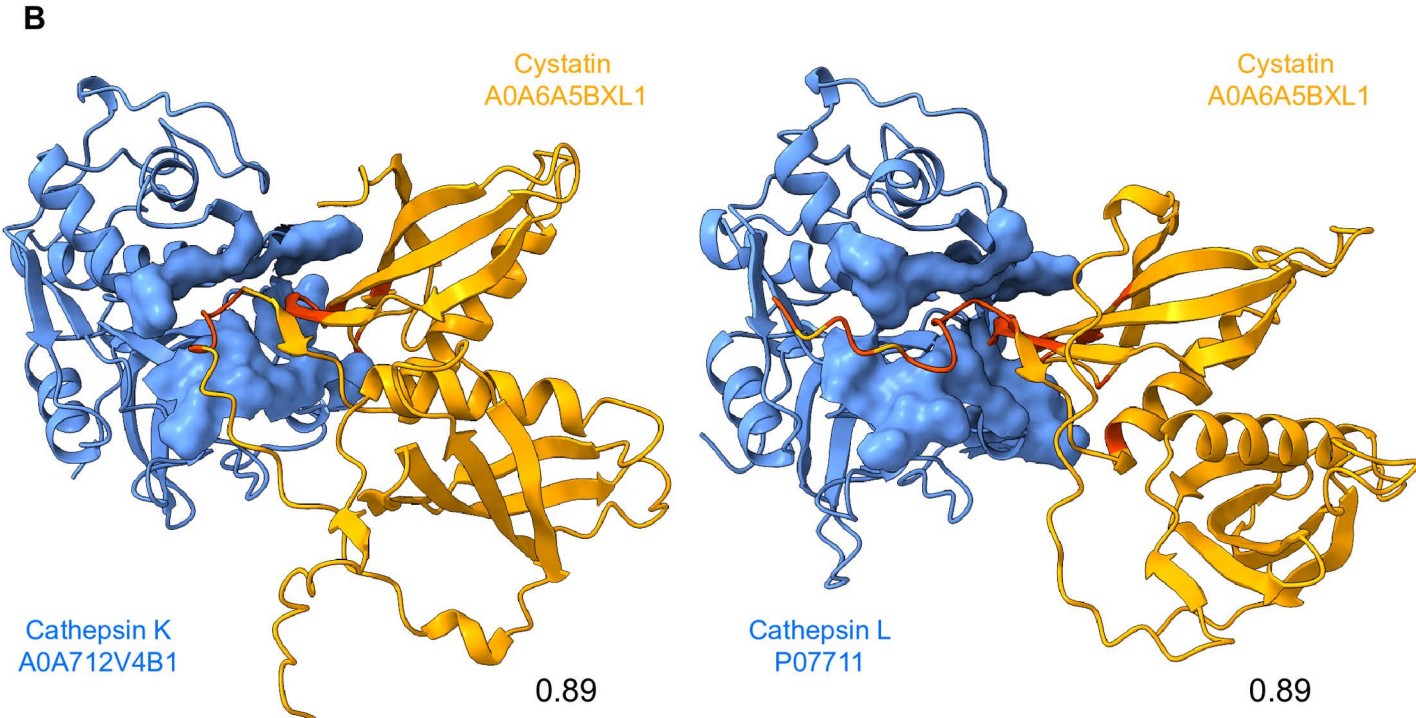

**Fig 3. Predicted interactions between *Naegleria fowleri* (Nf) cystatin and cysteine proteases from *Naegleria fowleri* and human genomes.** (A) Best structures predicted by AlphaFold2 multimer from 69 *N. fowleri* cysteine proteases with their ipTM scores. (B) Best structures predicted by AlphaFold2 multimer from 237 Homo sapiens cysteine proteases, with their ipTM scores. *N. fowleri* and *H. sapiens* cysteine proteases are shown in blue. Nf cystatin is shown in orange, with its binding site in red.

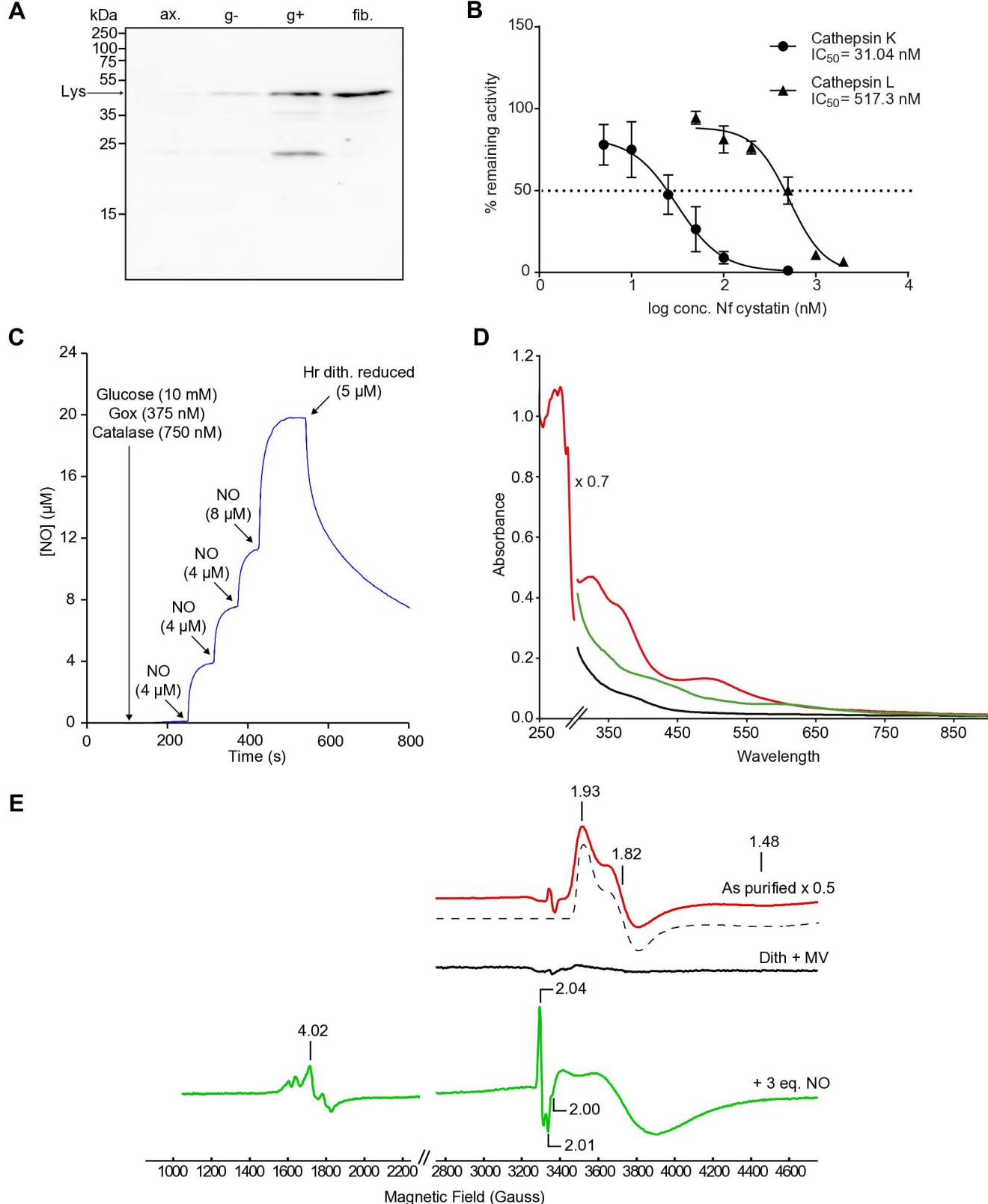

**Fig 4. Functional characterization of recombinant *Naegleria fowleri* (Nf) lysozyme, cystatin, and hemerythrin.** (A) Immunoblot detection of Nf lysozyme in cell lysates from axenically cultured *N. fowleri* (ax.) and amoebae co-cultured with *Klebsiella aerogenes* (g-), with *Micrococcus lysodeikticus* (g+), and with HT1080 fibrosarcoma cells (fib.) for 5 passages. Loading control is shown in S7B Fig. (B) Relative inhibition of human cathepsins K and L

by recombinant Nf cystatin. The dotted line indicates 50% inhibition of peptidase activity ($IC_{50}$). Relative inhibition values are related to enzyme activities with 0 nM Nf cystatin, which were taken as 100%. Enzymes were used at 50 nM concentrations. The $IC_{50}$ values were extrapolated using non-linear regression (variable slope, 4 parameters). Bars represent S.D. (C) UV-vis spectra of recombinant Nf hemerythrin. Black line shows the deoxy form of hemerythrin obtained after reduction with sodium dithionite and methyl viologen under anaerobic conditions. The red line shows the oxidized form of hemerythrin with absorption maxima at 330 nm, 380 nm and 500 nm representing the $O_2$-bound protein. The green line shows spectra after addition of 1 equivalent of nitric oxide (NO) to the fully reduced hemerythrin with a small band centered at 620 nm and other to absorption bands with maxima at 330 and 420 nm. (D) NO binding by hemerythrin measured by a modified Clark-type selective electrode showing four consecutive additions of NO stock solution to reach 20 μM, followed by the addition of 5 μM protein. (E) EPR spectra of as purified hemerythrin (red line), sodium dithionite and methyl viologen reduced hemerythrin (black line) and reduced hemerythrin incubated with 3 equivalents of nitric oxide (green line). Experimental conditions: protein concentration, 200 μM; temperature, 7 K; microwave frequency, 9.39 GHz; modulation amplitude, 1.0 mT; Microwave power, 2 mW. Spectral simulations and image preparation were performed using SpinCount [73].

virulence factors such as integrin-like proteins [8] and Mp2CL5 [33]. Nevertheless, paralog of Nfa1, hemerythrin (A0A6A5BXL4), was identified only in brain-isolated *N. fowleri*. This hemerythrin (here after named as Nf hemerythrin) belongs to the $O_2$-binding family [34] and may be able to bind nitric oxide, as was recently shown for hemerythrin-like protein from *Mycobacterium kansasii* [35,36], thus putatively playing a protective role as a NO scavenger.

## Characterization of putative virulence factors

Considering that Nf lysozyme is an *N. fowleri*-specific protein of unknown role, Nf cystatin is a potential inhibitor of host cysteine proteases, and Nf hemerythrin is a potential NO scavenger, we decided to further focus on these proteins and perform pilot experimental analyses. Recombinant Nf lysozyme and Nf cystatin with polyhistidine tag at the C-terminus were produced in *Escherichia coli* and purified by affinity chromatography on Ni-NTA agarose under native conditions. Recombinant Nf hemerythrin was also produced in *E. coli* but purified by two consecutive chromatographic steps to ensure correct folding of the protein and its diiron center.

Analytical size exclusion chromatography was used to determine the native molecular mass of all three proteins in solution. The dominant peak for recombinant Nf lysozyme (S6 Fig) corresponded to ≈ 203 kDa, indicating a pentameric structure. Two identified peaks of the recombinant Nf cystatin (S6 Fig) corresponded to ≈ 126 kDa and ≈ 60 kDa, suggesting tetramer and dimer formation, respectively, in comparison to previously described cystatins from *N. fowleri* that showed monomeric and dimeric structures [30,31]. The dominant peak for recombinant Nf hemerythrin (S6 Fig) corresponded to ≈ 15 kDa, showing a monomeric form similar to previously described for the hemerythrin from *N. gruberi* [34].

An assay using *Micrococcus lysodeikticus* bacteria was used to test the activity of recombinant Nf lysozyme by monitoring bacterial lysis with a UV-visible spectrophotometer. There was no change in absorbance at 450 nm upon addition of the enzyme from *N. fowleri*, whereas chicken egg white lysozyme effectively lysed the bacteria, resulting in a decrease in absorbance over time (S7A Fig). To study the induction of Nf lysozyme upon interaction with both gram-negative and gram-positive bacteria and mammalian cells, we performed immunoblot analysis of *N. fowleri* cells long-term co-cultured with *K. aerogenes* (gram-negative), *M. lysodeikticus* (gram-positive) and HT1080 fibrosarcoma cells (Fig 4A) using polyclonal antibody raised against recombinant protein. Immunoblots showed an increase in the intensity of antibody recognition of Nf lysozyme, suggesting an active involvement of this protein in the interaction with some bacteria and with mammalian cells, which could have an impact on the infection of the host.

Recombinant Nf cystatin was used to measure its inhibitory capacity towards different cysteine proteases. It did not inhibit bovine cathepsin B at any of the concentrations used. However, it proved to be a potent inhibitor of human cathepsin K ($IC_{50}$ ≈ 31 nM), while $IC_{50}$ with human cathepsin L was more than an order of magnitude higher (≈ 517 nM) (Fig 4B).

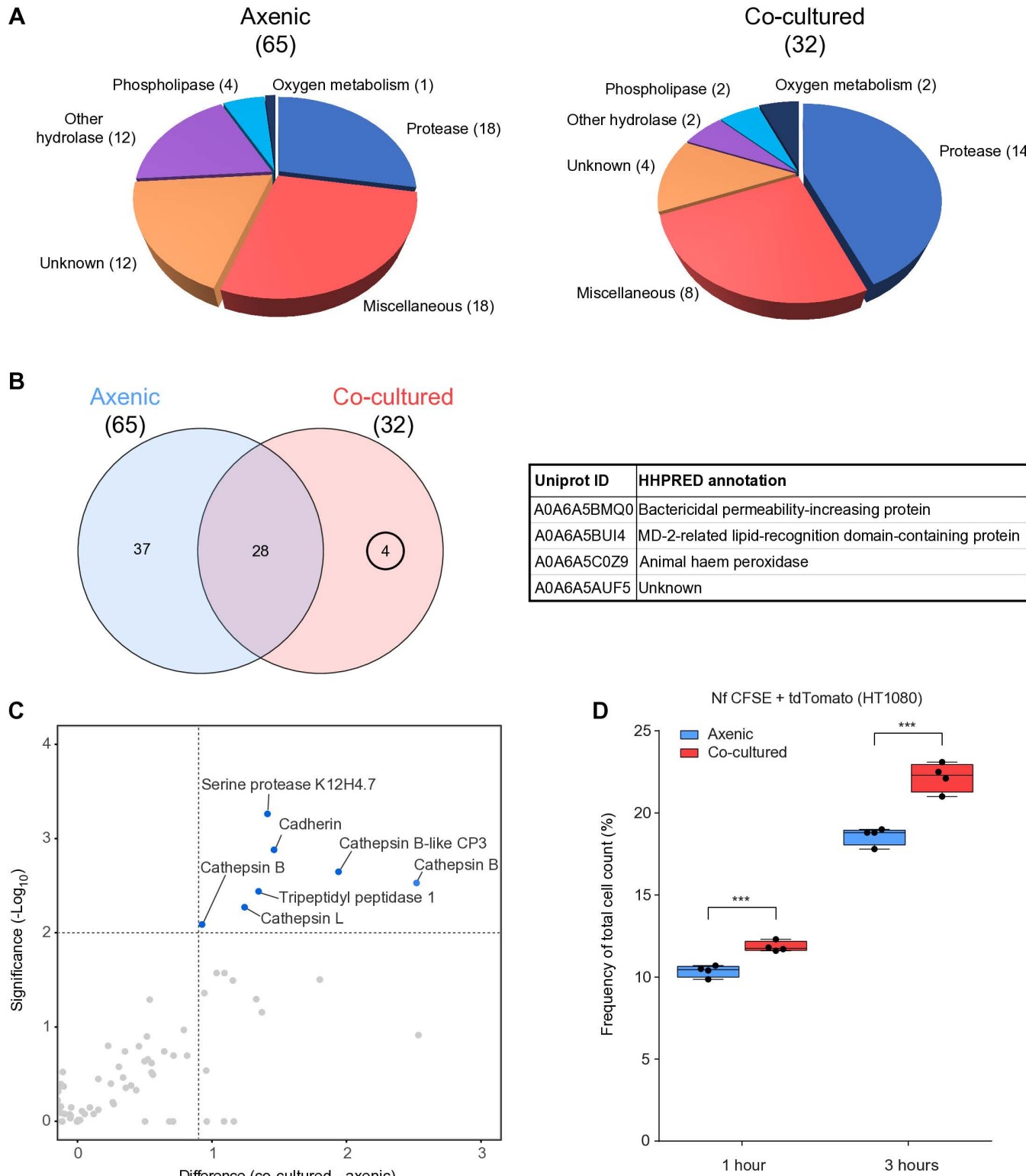

**Fig 5. Long-term co-culture of *Naegleria fowleri* with mammalian cells affects amoeba-secreted proteins.** (A) Axenic *N. fowleri* [65] and long-term co-cultured *N. fowleri* [32] proteins identified in their respective secretomes, sorted into 6 functional categories using Pfam motif identification and HHpred annotation. (B) Venn diagram showing the overlap of axenic *N. fowleri* and long-term co-cultured *N. fowleri* secretomes after incubation in an axenic medium. Four proteins identified only in the co-cultured amoeba secretome are listed in the table. (C) Volcano plot depicting significantly enriched

proteins in the secretome of long-term co-cultured amoebae compared to the secretome of axenic amoebae. Both secretomes were prepared after one hour of incubation of amoebae with HT1080. (D) Graph showing changes in the percentage of CFSE-labeled *Naegleria* with ingested tdTomato from HT1080 cells after 1 and 3 hours of incubation of fibrosarcoma cells with either axenic amoebae or long-term co-cultured amoebae as measured by flow cytometry. (*** p-value<0.001).

To investigate the potential binding of nitric oxide to recombinant Nf hemerythrin, the protein was initially characterized spectroscopically by UV-visible and Electron Paramagnetic Resonance (EPR). The UV-visible spectra of the as prepared Nf hemerythrin showed four major absorption bands with maxima at 280 and 330, 380 and 500 nm, where the last three are characteristic of the oxy-hemerythrin form, reflecting the presence of the covalent HOO–Fe(III) and Fe(III)–μO–Fe(III) bonds (Fig 4C – red line) [37,38]. After anaerobic reduction with stoichiometric amounts of sodium dithionite and methyl viologen, the reduced protein showed no significant absorption bands, which is consistent with the formation of the deoxy-hemerythrin form (Fig 4C - black line) [37,38]. This form was utilized to evaluate the ability of this protein to bind NO by incubating it, anaerobically, with an NO-saturated buffer solution, and monitoring the reaction by UV-visible. The incubation of the reduced Nf deoxy-hemerythrin with one equivalent of NO led to the formation of two broad absorption bands with maxima at ~420 and 620 nm (Fig 4C – green line). These absorption bands are similar to the ones observed previously for the *M. kansasii* hemerythrin-like protein (HLP) and are indicative of the formation of a mononitrosyl–iron species, {FeNO}$^7$ (Enemark–Feltham notation) [36].

The NO binding by Nf deoxy-hemerythrin was also determined amperometrically with a modified Clark-type electrode, selective for NO (Fig 4D). The fully reduced Nf hemerythrin reacted with the NO in solution and a consumption of c.a. three NO molecules per protein was verified. This observation was again in agreement with what was observed by Albert and co-workers for the *M. kansasii* HLP, which indicates that the Nf hemerythrin follows a similar NO binding mechanism [36,39]. The EPR spectra of the as prepared Nf hemerythrin sample presented a rhombic signal with g values 1.48, 1.82 and 1.93, which are characteristic of an antiferromagnetically coupled mixed-valence Fe(II)–Fe(III) center with $S=1$/2 (Fig 4E – red line and black dashed line - simulation), similar to the described for other hemerythrins and also other diiron containing proteins [35,40]. Upon reduction with sodium dithionite and methyl viologen, the EPR signal was no longer observed, which is consistent with the formation of an all-ferrous Fe(II)–Fe(II) oxidation state with a total spin system of S=0 (Fig 4E - black line). When the reduced Nf hemerythrin (same sample monitored by UV-vis) was incubated with three equivalents of NO, a EPR signals with resonances centered around g ~ 2 and 4, which are indicative of the formation of a low-spin $S=1$/2 and high-spin $S=3$/2 iron-nitrosyl species, {FeNO}$^7$ (Fig 4E – green line) [36].

The structure of Nf hemerythrin, including the iron atoms of the diiron catalytic center, was predicted with AlphaFold3 [41]. The result shows a 4-helix bundle type of structure, similar to other hemerythrins from diverse organisms, from worms (*Themiste hennahi*, PDB 2MHR or *Themiste dyscritum*, PDB 1HMD) to bacterium (*Methylococcus capsulatus*, PDB 4XPX) [42–44]. The coordination of the diiron center is also similar to other hemerythrins, with the hexa-coordinated Fe1 and penta-coordinated Fe2 bound to (Nf hemerythrin numbering) His75, 79 and 107 (Fe1) and His26 and 56 (Fe2) and also by the bridging Glu60 and Asp112. The presence of a μ-oxo bridge is predicted to complete the coordination of both atoms (S8 Fig). This leaves the Fe2 with one free position to bind the O$_2$ or NO molecule.

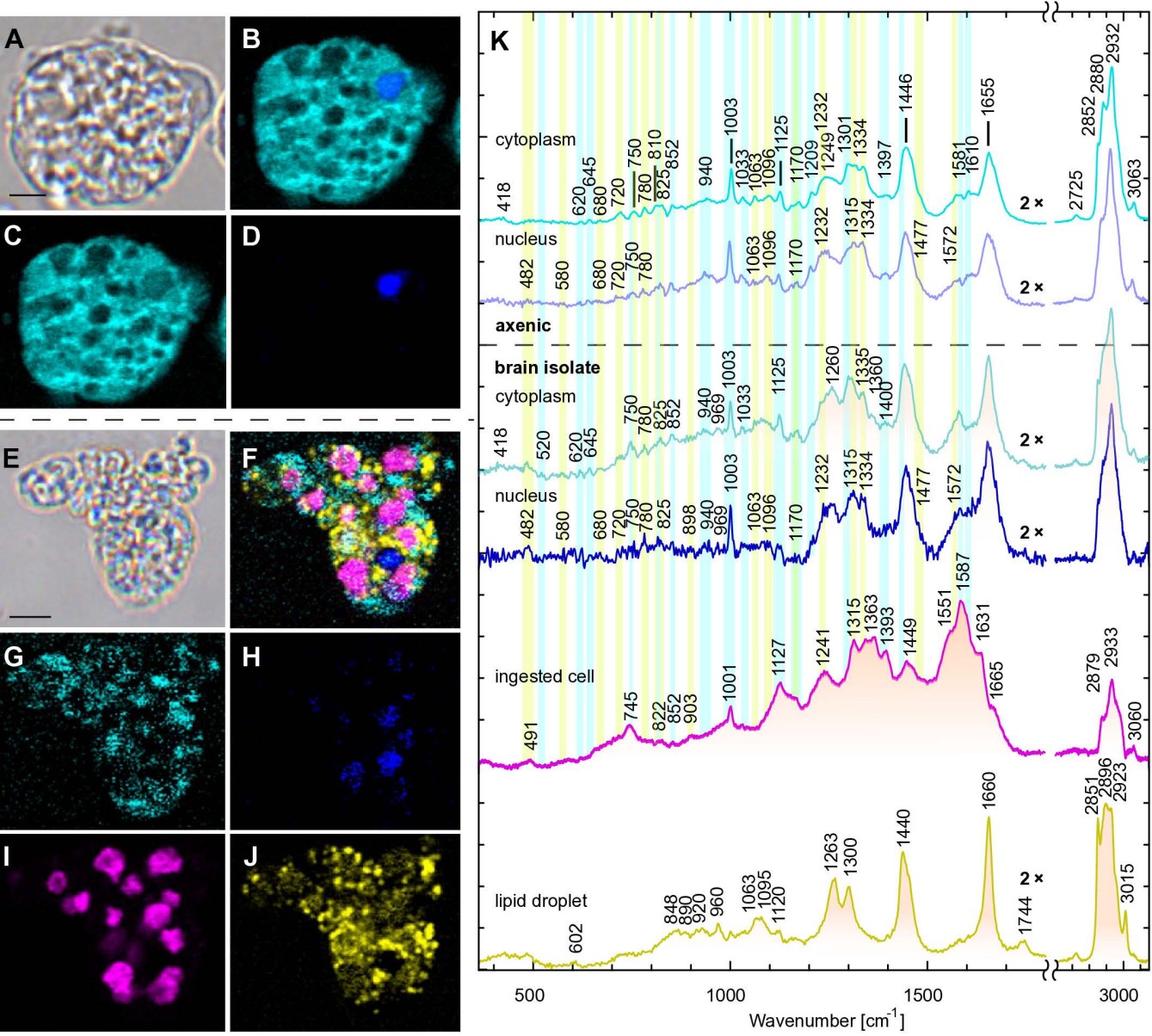

**Fig 6. Raman chemical maps of *Naegleria fowleri* grown axenically (A–D) and isolated from mouse brain (E–J).** A – bright field, B – merge, C – cytoplasm, D – nucleus, E – bright field, F – merge, G – cytoplasm, H – nucleus, I – ingested cells, J – lipid droplets, scalebar 5 μm; K – decomposed Raman spectra of axenic cell culture in the upper part and cells isolated from mouse brain in the lower part (shaded in orange). The intensities are normalized to the maximum value with an exception for the fingerprint region below 1800 cm⁻¹ that has been doubled (marked as 2×) for better display of the spectra compared to the more intense high-wavenumber region above 2700 cm⁻¹. However, the spectra of the ingested cells show an opposite trend, their spectrum has not been adjusted. Peak wavenumbers are displayed separately for four significantly different spectral clusters: cytoplasm (cyan), nucleus (blue), ingested cells (magenta), and lipid droplets (yellow). The characteristic peaks of amino acids and protein-related signals are highlighted by blue lines, nucleosides-, nucleotides- and nucleic acid-related components in yellow.

## Long-term co-culture of *N. fowleri* with mammalian cells increases amoeba cytopathogenicity but does not affect trogocytosis

The contact-independent cytopathogenicity of microorganisms is largely based on the secretion of virulence factors. Therefore, in addition to cellular proteomics, we focused our study on extracellular proteins secreted by amoebae. The axenically maintained *Naegleria* and

*Naegleria* long-term co-cultured with human fibrosarcoma cells were incubated in a protein-free medium for 1 h and the secretomes were analyzed by proteomics. A total of 200 proteins in axenically maintained amoebae and 105 proteins in long-term co-cultured amoebae were identified. To distinguish secreted proteins from contaminants, we used a combination of prediction software for subcellular localization of eukaryotic proteins: DeepLoc 2.0, SecretomeP 2.0, SignalP 6.0 and TargetP 2.0. The analyzed protein was considered to be secreted if it had a positive prediction with at least three of the four prediction software and was identified in at least three out of four proteomic samples for a given condition. After applying these criteria, we obtained a list of 65 proteins secreted by axenically maintained *N. fowleri* and 32 proteins secreted by *N. fowleri* after long-term co-culturing with human fibrosarcoma cells (S5 Table). These proteins were manually sorted into 6 functional categories considering Pfam motif identifications and HHpred searches (Fig 5A). We identified 28 proteins shared between the two secretomes, 37 proteins identified only from axenically maintained amoebae and 4 proteins identified only from co-cultured amoebae, representing the candidate virulence factors (Fig 5B). Two proteins identified only after long-term co-culturing with host cells, bactericidal permeability-increasing protein (A0A6A5BMQ0) and MD-2-related lipid-recognition domain-containing protein (A0A6A5BUI4), are lipid-binding proteins that may be involved in the recognition of prey cells and their associated products.

To obtain a more detailed and relevant set of secreted proteins potentially involved in *N. fowleri* virulence, we examined the differences in protein secretion between axenically cultured amoebae and those long-term co-cultured with host cells, after one hour of incubation with fibrosarcoma cells. Although this setup is experimentally more challenging due to host protein contamination, it also allows the identification of proteins the secretion of which depends on the presence of host cells. We identified 7 proteins with significantly higher secretion from long-term co-cultured amoebae (Fig 5C and S6 Table), 4 of which are cysteine proteases: cathepsin B (A0A6A5C7D7), cathepsin B-like CP3 (A0A6A5CEE5), cathepsin L (A0A6A5BXT3) and cathepsin B (A0A6A5C6R6), and 2 of which are serine proteases: serine protease K12H4.7 (A0A6A5CEZ9) and tripeptidyl peptidase 1 (A0A6A5BSJ3). The seventh identified protein, cadherin (A0A6A5BLV3), may play a role in cell adhesion.

It has long been known that long-term co-culture of *N. fowleri* with mammalian cells results in increased virulence [20]. However, it is unclear whether the increase in virulence is due to faster cell destruction by trogocytosis or increased secretion of proteases or other molecules capable of cell lysis. In our study, we used the flow cytometry method illustrated in Fig 1A and compared the percentage of CFSE-labeled amoebae containing tdTomato fluorescent protein from HT1080 fibrosarcoma cells after 1 and 3 hours of co-culture with either axenically maintained *N. fowleri* or long-term co-cultured *N. fowleri*. After 1 hour of incubation, we observed higher levels of ingested tdTomato in long-term co-cultured amoebae compared to axenic amoebae, which increased over time as seen after 3 hours of incubation (Fig 5D, representative flow cytograms in S9 Fig). In order to ascertain whether long-term co-cultured amoebae exhibited enhanced efficacy in the destruction of host cells, rather than merely adapting to the conditions and accelerating their growth, we calculated doubling times for each condition in host cell-free medium and observed no difference in their proliferative rates (S10A Fig). To further elucidate the mechanism of contact-dependent host cell lysis, we performed the flow cytometry experiment with CellTrace Far Red-labeled axenic or long-term co-cultured *N. fowleri* after incubation with HT1080 fibrosarcoma cells labeled with either cytosolic eGFP or membrane-targeted eGFP-CAAX. After 3 hours of co-culture, we observed higher levels of ingested cytosolic eGFP in long-term co-cultured N. fowleri (S10B Fig), similar to the results shown in Fig 5D. However, there was no difference in the number of amoebae that ingested membrane-targeted eGFP-CAAX between axenic or long-term co-cultured

*N. fowleri* (S10C Fig), indicating that the higher rate of host cell ingestion is most likely not due to trogocytosis.

## Raman microspectroscopy highlights the presence of lipid stores in *Naegleria* isolated from brain

Using Raman microspectroscopy, a label-free chemical imaging technique, we analyzed the cell composition of *N. fowleri* from two different settings – the axenically grown amoebae and those that were extracted from the mouse brain (Fig 6). Their cytoplasm showed protein and nucleotide content (Fig 6K, highlighted by blue and yellow lines, respectively). The spectra are dominated by the signal from the protein backbone, namely, by the amide I, the broad band at $1655 \text{ cm}^{-1}$, reflecting the protein secondary structure content (with vibrations of α-helices at lower wavenumbers and β-sheets/β-turns at higher wavenumbers) and the amide III region $\sim 1225–1310 \text{ cm}^{-1}$ (where lower wavenumbers correspond to the signal of β-sheets whereas α-helices lies at higher wavenumbers). Another strong broad band at $1446 \text{ cm}^{-1}$ mostly corresponds to bending vibrations of the $CH_2$ and $CH_3$ groups [45]. Other narrow bands mostly reflect vibrations of amino acids with aromatic side chains, such as phenylalanine twisting, ring stretching, and deformations (620, 1003, $1033 \text{ cm}^{-1}$, respectively) and tyrosine twisting, Fermi resonance doublet, bending, and deformations (645, 825 and 852, 1170, $1610 \text{ cm}^{-1}$, respectively), and tryptophane (750, 1334, and $1581 \text{ cm}^{-1}$) with less-specific vibrational modes of aromatic amino acids (1209, 1581, 1600–1620, $3063 \text{ cm}^{-1}$) [45–47]. If we compared the Raman signal of cytoplasm between the axenically grown and brain-isolated cells, the latter showed lower content of aromatic amino acids (1003, 1335/$1360 \text{ cm}^{-1}$) for data normalized for amide I band. Additionally, Raman spectra of cytoplasm contained signal of nucleotides and nucleic acids, mostly RNA, and potentially DNA (Fig 6K yellow bands) from mitochondria. Their major components were symmetric and asymmetric O–P–O phosphodiester stretching vibrations (580, 780, 825, 898, 1063, 1096, $1232 \text{ cm}^{-1}$), nucleotides (720, 780, 1170, 1315, 1334, $1477 \text{ cm}^{-1}$), and RNA specifically (810, $1232 \text{ cm}^{-1}$). The Raman fingerprints of the cell nucleus under both growth conditions have comparable profiles and intensities of proteins (as listed above) and some additional signal of DNA and nucleic acids in general (482, 680, 720, 750, 1096, 1170, 1232, 1315, 1334, 1477, $1572 \text{ cm}^{-1}$) [45,47]. These complex spectral fingerprints tend to overlap, e.g., the positions assigned to the phosphate and $NH_2$ groups have similar positions with nucleotides, nucleic acids, some amino acids and also amide III [48–50].

The most significant difference observed in cells isolated from the mouse brain was remnants of undigested phagocytized cells that were visible even in the transmission light. Those parts were spectrally distinct from the amoeba cytoplasm, showed an increased fluorescence background, and due to their hemoglobin contents resembled red blood cells in the process of digestion (Fig 6I and 6K, purple line). Our scanning 532 nm laser resonantly enhanced the significant peaks in hemoglobin spectra (745, 1127, 1315, and $1587 \text{ cm}^{-1}$) [51]. The complex of peaks 1340–1393 and with the ratio of 1587 and $1551 \text{ cm}^{-1}$ pointed at the high proportion of oxidized iron $Fe^{3+}$ levels [47].

Finally, unlike axenically grown cells, cells isolated from the mouse brain contained dozens of lipid droplets (Fig 6J and 6K, yellow line). Their composition reflected Raman vibration spectra of triacylglycerols: the ester bond with C=O stretching ($1744 \text{ cm}^{-1}$) and the mixture of variable fatty acid derivatives represented by $CH_2$ and $CH_3$ deformations, stretching, twisting and scissoring (890, 920, 960, 1063, 1095, 1120, 1300, $1440 \text{ cm}^{-1}$) with unsaturated double bond =C–H deformation ($1263 \text{ cm}^{-1}$) and the C=C stretching ($1660 \text{ cm}^{-1}$). Additionally, we analyzed the level of lipid unsaturation, expressed as the ratio of C=C/$CH_2$ vibrations (1660 and $1440 \text{ cm}^{-1}$, respectively), yielding the ratio of $1.39 \pm 0.17$ ($N = 10$), which exceeded the average count of two double bonds per fatty acid, based on the calibration of standard fatty acids [52].

## Discussion

Since the discovery of *N. fowleri* as the causative agent of primary amoebic encephalitis, there has been debate about why it is pathogenic compared to other *Naegleria* species. For several decades, many potential virulence factors have been described [16–18,21], but a definitive conclusion on the pathogenicity of the amoebae has yet to be reached.

The objective of our study was to gain a comprehensive insight into the interactions between *N. fowleri* and its prey and to identify novel virulence factors.

Our whole-cell label-free comparative proteomic analysis demonstrated a markedly reduced response of *Naegleria* to its host cells after short-term co-culturing compared to long-term co-culture. Consequently, our attention was focused on the upregulated proteins that were shared between the long-term fibrosarcoma co-culture and the brain-isolated amoebae. This resulted in the identification of a set of proteins involved in sulfur and lipid metabolism, as well as a protein predicted to be lysozyme, which had been shown to be *N. fowleri*-specific [21]. Although recombinant Nf lysozyme did not exhibit the activity of typical lysozymes, the expression of the protein was induced upon long-term interaction with either bacteria or human cells, suggesting its involvement in the interaction with the prey. The lack of activity may be attributed to improper protein folding, as the recombinant protein contains disordered regions that require interaction with an appropriate partner for optimal functionality. Given the above, we believe this protein may represent a key player unique to *N. fowleri* prey cell killing.

Examination of the proteome of brain-isolated amoebae revealed the presence of two additional proteins that were exclusively upregulated in this condition. One of these proteins, Nf cystatin, belongs to a group of proteins that act as cysteine protease inhibitors [30,31], therefore it may play an essential role in the interaction of *N. fowleri* with host immune cells. It has also been shown that other cystatins from *N. fowleri* are capable of inhibiting endogenous cysteine proteases [30,31]. Our bioinformatic prediction of interactions between Nf cystatin and cysteine proteases showed the highest interaction scores among the human cysteine proteases with cathepsin K and cathepsin L and the inhibition of these cathepsins was confirmed experimentally, while the recombinant Nf cystatin showed no inhibition of cathepsin B. Both cathepsin L and K may be found in the central nervous system where they are capable of cleaving neuroactive peptides [53,54]. Cathepsin K is also able to cleave bradykinin and other kinins [55] which have pro-inflammatory effects. Cathepsin L has an increased secretion from microglia in response to lipopolysaccharides, supporting its role in contributing to inflammatory responses [56]. Our results also contrast with those of previously studied *N. fowleri* cystatins (A0A6C0TC17, A0A6A5BTM4), which mainly inhibited cathepsin B [30,31], except for fowlerstefin, which also inhibited human cathepsin L [30]. Recent proteomic characterization of *N. fowleri* extracellular vesicles identified Nf cystatin within these vesicles [19]. Therefore, Nf cystatin appears to be secreted, but its role in defense against host proteases should be further investigated.

Another exclusively upregulated protein in the proteome of brain-isolated amoebae, hemerythrin, is a paralog of the major *N. fowleri* virulence factor Nfa1 [32,34]. Nf hemerythrin is a member of the $O_2$-binding family [34] and may therefore be able to bind nitric oxide. Indeed, our biochemical and biophysical analyses clearly demonstrated its ability to bind NO. The nitric oxide binding mechanism observed for Nf hemerythrin appears to be similar to the hemerythrin-like protein from the bacterium *M. kansasii*, which has the ability to consume more than one equivalent of NO compared to deoxyhemoglobin [36]. Nitric oxide has been demonstrated to contribute to the amoebicidal activity of macrophages against *N. fowleri* [57]. Consequently, hemerythrin may facilitate the amoeba's ability to cope with elevated levels of NO produced by macrophages, thereby allowing some evasion from host defenses.

Comparative analyses of the secretomes of amoebae maintained under two distinct conditions, axenically and following long-term co-culture with host cells, identified two lipid-binding proteins, bactericidal permeability-increasing protein and MD-2-related lipid-recognition domain-containing protein, which were secreted only by long-term co-cultured amoebae. Bactericidal permeability-increasing protein is a highly expressed protein in human neutrophils that binds lipopolysaccharides of Gram-negative bacteria [58]. This protein has also been shown to be essential for efficient phagocytosis by neutrophils [59]. Concurrently, *Entamoeba histolytica* EhNPC2 proteins, which contain the MD-2-related lipid recognition domain, are involved in exogenous cholesterol uptake and cellular trafficking, with implications for motility and phagocytosis [60]. Consequently, these two proteins may contribute to the efficiency of cytopathogenicity in *N. fowleri*.

To gain a more detailed insight into the secreted proteins potentially involved in the interactions with the host, we performed a comparative analysis of protein secretion between axenically cultured amoebae and those long-term co-cultured with host cells after short incubation with fibrosarcoma cells. Under these conditions, we identified 6 proteases, including two cathepsins B, cathepsin B-like CP3, cathepsin L, serine protease K12H4.7 and tripeptidyl peptidase 1, and protein cadherin, which may play a role in cell adhesion, to be upregulated under long-term co-culture with host cells. Previous studies have implicated cathepsins B as a virulence factor during *N. fowleri* infection and their ability to induce inflammatory responses in BV-2 microglial cells [61,62]. Cathepsin B-like CP3 was also found to be upregulated on RNA level in mouse-passaged *N. fowleri* and has no ortholog in *N. gruberi*, raising the possibility of its involvement in pathogenesis [21]. The remaining upregulated secreted proteases have also been demonstrated to be upregulated in mouse-passaged *Naegleria* [21]. In addition, three of the proteases from the secretomes obtained in our study, cathepsin B, F and serine protease K12H4.7, have been identified in *N. fowleri* extracellular vesicles in a recent study [19]. This suggests that some proteins may be secreted from the amoeba both directly and through extracellular vesicles.

Following the secretome analysis showing an increase in protease secretion in long-term co-cultured *N. fowleri*, we sought to ascertain whether a similar change in trogocytosis would be observed between axenic and co-cultured amoebae. While it has long been observed that co-culture of *N. fowleri* with mammalian cells increases virulence [20], the mechanism behind this phenomenon was unclear. Our flow cytometry analysis revealed that while long-term co-cultured amoebae exhibited a higher rate of fibrosarcoma cells ingestion over time than axenically maintained amoebae, this increase is most probably not due to trogocytosis. We therefore propose that the increased virulence of *N. fowleri* is mainly due to the increased secretion of proteases capable of cell lysis, while the rate of trogocytosis remains constant.

The application of Raman microspectroscopy revealed the existence of markedly disparate patterns between the axenically cultivated amoebae and those extracted from the mouse brain. The axenically grown cells exhibited a multitude of empty vacuoles, whereas those from the mouse brain displayed lipid droplets and vacuoles containing ingested host cells. The higher content of nucleic acids and hemoproteins indicates a higher metabolic activity of the latter. The Raman signal of the amoeba cytoplasm is rather complex and of low intensity, consisting mainly of water with a complex mixture of ions, proteins, nucleic acids, and phospholipid-based endomembrane systems. Additionally, it contains a diffuse signal from mitochondria that cannot be spectrally or spatially separated from the chemical maps. This complex mixture of biomolecules exhibits patterns similar to those reported in the spectra of other unicellular or multicellular heterotrophs and extracted biopolymers [50,63,64]. As previously stated, in contrast to axenically grown cells, amoebae isolated from mouse brain tissue exhibited the presence of numerous lipid droplets. This finding is consistent with the Raman data from

healthy brain tissue, which is known to contain high levels of polyunsaturated fatty acids [50]. This also indicates that amoebae consume brain cells and store the lipids they ingest. As previously demonstrated in *N. gruberi*, a nonpathogenic relative of *N. fowleri*, β-oxidation of lipids functions as primary energy metabolism, an unprecedented metabolic feature among protists grown under aerobic conditions compared to the canonically preferred glucose utilization [65]. The same is predicted for *N. fowleri*, in which our study showed that lipids can be stored at high levels in brain-eating cells compared to axenically grown cells.

In summary, our work has provided a comprehensive view of *N. fowleri* interactions with the host/prey and outlined several unique components of the machinery responsible for its cytopathogenicity. These are important potential targets for antiparasitic intervention but are also of great interest in understanding how a free-living amoeba becomes a successful human pathogen.

## Materials and methods

### Ethics statement

Animal procedures were approved by the Czech Ministry of Agriculture (53659/2019-MZE-18134). The study protocol was approved by the Ethics and Animal Welfare Committee of the Charles University (Prague, Czech Republic) no. MSMT-37682/2019-3.

### Cultivation

*Naegleria fowleri* strain HB-1 was maintained in 2% Bacto-Casitone (Difco, USA) supplemented with 10% heat-inactivated fetal bovine serum (Thermo Fisher Scientific, USA), with the addition of penicillin (100 U/ml) and streptomycin (100 μg/ml) at 37 °C. HT1080 fibrosarcoma cells (with overexpressed tdTomato, eGFP or eGFP-CAAX protein), kindly provided by dr. Daniel Rösel (Department of Cell Biology, Charles University, Prague, Czech Republic), were cultured in Dulbecco's modified Eagle's medium (Sigma-Aldrich, USA) supplemented with 10% heat-inactivated fetal bovine serum (Thermo Fisher Scientific), with the addition of penicillin (100 U/ml) and streptomycin (100 μg/ml) at 37 °C. For co-culture experiments of *N. fowleri* with HT1080 cells, a mixture of Bacto-Casitone medium and Dulbecco's modified Eagle's medium was used in a 1:1 ratio. Long-term co-cultures are defined as *N. fowleri* cells that have undergone at least five continuous passages with HT1080 fibrosarcoma cells.

### Mice infection

Female BALB/c mice were infected intranasally with $2 \times 10^4$ *N. fowleri* cells in 30 μL phosphate-buffered saline (PBS) under diethyl ether anesthesia. *N. fowleri* cells were passaged through mice two times prior to experimental infections to ensure a high and stable infection rate. Amoeba cells were isolated directly from the brains of euthanized mice after 5-days of infection and placed into growth medium in culture flasks at 37°C until a sufficient number of parasites had migrated from the brain (≈2 hours). These amoebae were used directly for further experiments.

### Flow cytometry

Prior to each co-culture experiment, *N. fowleri* cells were labeled with 5 μM carboxyfluorescein succinimidyl ester (CFSE, Thermo Fisher Scientific) or 1 μM CellTrace Far Red dye (Thermo Fisher Scientific) in PBS for 20 min at 37 °C, followed by incubation in the fivefold volume of Bacto-Casitone medium for 5 min. For experiments investigating the effect of actin inhibition, CFSE-labeled amoeba cells were preincubated with 50 μM CK-666, 10 μM

wortmannin, 100 µM piceatannol, and 10 µM PP2 for 30 min, then mixed with HT1080 fibrosarcoma cells containing tdTomato fluorescent protein in a 1:1 ratio (amoeba:fibrosarcoma) and incubated at 37 °C for 3 hours. After incubation, the co-cultures were analyzed using a Guava easyCyte 8HT flow cytometer (Luminex Corporation, USA) with a 488 nm laser (150 mW) for excitation and Red-B (BP695/50) and Green-B (BP525/30) emission filters corresponding to red and green fluorescence parameters. As controls, *N. fowleri* and HT1080 were incubated individually with the same concentrations of each inhibitor for 3 hours and analyzed by flow cytometer. In addition, amoeba cells were monitored throughout the incubation to ensure that no flagellate cells were formed. For experiments investigating the difference between axenically cultured *Naegleria* and long-term co-cultured *Naegleria*, CFSE-labeled or CellTrace Far Red-labeled amoebae were mixed with HT1080 fibrosarcoma cells containing tdTomato fluorescent protein or eGFP, eGFP-CAAX fluorescent protein in a 1:1 ratio (amoeba:fibrosarcoma) and incubated at 37 °C for 1 and 3 hours and analyzed using a Guava easyCyte 8HT flow cytometer (Luminex Corporation) with a 488 nm laser (150 mW) for excitation and Red-B (BP695/50) and Green-B (BP525/30) emission filters corresponding to the red and green fluorescence parameters for tdTomato, CFSE, eGFP and eGFP-CAAX. For CellTrace Far Red, a 642 nm laser (100 mW) was used for excitation and a Red-R (BP661/15) emission filter was used as the red fluorescence parameter. All data were analyzed using FlowJo v10 software (BD Biosciences).

### Live-cell imaging

As in the flow cytometry experiments, *N. fowleri* cells were labeled with 5 µM CFSE or 1 µM CellTrace Far Red, mixed with HT1080 cells in 35 mm glass bottom dish (Cellvis, USA) and incubated at 37 °C for 1 hour. Prior to imaging the culture medium was changed to glucose medium (50 mM glucose, 0.5 mM $MgCl_2$, 0.3 mM $CaCl_2$, 5.1 mM $KH_2PO_4$, 3 mM $Na_2HPO_4$, pH 7.4) to reduce the non-specific signal. Samples were imaged with a Nikon CSU-W1 spinning disk field scanning confocal microscope (Nikon Corporation, Japan) using a CFI Plan Apo VC 60× C/1.20 [N.A.] water-immersion objective. Excitation wavelengths were 405, 488, 561 and 638 nm, while emission filters were gated to 430–475, 500–550, 575–625 and 608–683 nm for the Hoechst 33342, CFSE, eGFP, eGFP-CAAX, tdTomato and CellTrace Far Red channels, respectively. Images were processed using ImageJ 1.52 [66]. For video acquisition, unlabeled *N. fowleri* cells were mixed with HT1080 cells in a µ-Dish 35 mm, low (ibidi GmbH, Germany) in a 1:1 ratio and observed on a Nanolive CX-A microscope (Nanolive SA, Switzerland) at 37 °C for a period of 6 hours.

### LC–MS

Label-free whole-cell comparative proteomic analysis was performed in independent biological triplicates of *N. fowleri* cultured under four different conditions: axenic, short-term 6 h co-culture with *K. aerogenes*, short-term 6 h co-culture with HT1080 fibrosarcoma cells and long-term co-culture (5 consecutive passages) with HT1080 fibrosarcoma cells. As a fifth condition, postmortem brains from three *N. fowleri*-infected mice were used. Brains were washed in growth medium in culture flasks at 37 °C until a sufficient number of parasites had migrated from the brain (≈ 2 hours). Cells from all conditions were washed three times in PBS (1200 × g, 10 min, 4 °C) and the dry pellet was subjected to proteomic analysis. For each condition, the method was performed as described in [67] using nanoflow liquid chromatography (LC) coupled to mass spectrometry (MS). The resulting data were evaluated using MaxQuant software [68] and searched against the AmoebaDB *N. fowleri* database, *K. aerogenes*, *Homo sapiens* (human), *Mus musculus* (mouse) Uniprot databases. Further processing

was performed with Perseus software [69] and selected proteins were manually annotated with HHpred [70]. Student's t-test with Benjamini-Hochberg correction was used to evaluate the significantly changed proteins at the level of a false discovery rate of 5%. The percentage of *N. fowleri* proteins identified in each condition was 90.2 ± 0.6% (NF-axenic), 89.9 ± 0.7% (NF+bacteria), 83.3 ± 1.5% (NF+HT1080-short-term), 89.0 ± 1.8% (NF+HT1080-long-term) and 78.2 ± 2.0% (NF-brain-isolate). Detailed characteristics of the proteomic analysis are summarized in S7 Table. The mass spectrometry proteomics data have been deposited to the ProteomeXchange Consortium via the PRIDE [71] partner repository with the dataset identifier PXD056622.

## Interactions and structure prediction by Alphafold

Custom Google colab script was used to analyze hypothetical interactions between Nf cystatin and cysteine proteases from *N. fowleri* and human. The sequences for cysteine proteases were downloaded from UniProtKb version 2023_05. The HHpred was deployed in order to identify and subsequently trim the N-terminal signal sequence and activation peptide since it blocks the binding region necessary for the interaction with Nf cystatin. The searches were conducted against the Pfam database version 36. AlphaFold2 multimer v.2 model was used for the predictions. Each prediction was done three times and ipTM values as well as binding residues from the three models were compared (S3 and S4 Tables). The script can be freely accessed at the following GitHub repository: https://github.com/vitdohnalek/Batch_AlphaFold2_multimer_v2.

The three-dimensional structure model of Nf hemerythrin was predicted using AlphaFold3 [41]. Structural analysis, visualization and images were performed using UCSF ChimeraX 1.8 [72].

## Gene cloning, expression and purification

Genes for Nf lysozyme (A0A6A5BDE0; primers: forward 5'-CACCATATGTCGTCTTACTA CACAAAGGAAC-3', reverse 5'-CACCTCGAGAGCATCGTCGTCGTCGTT-3'), Nf cystatin (A0A6A5BXL1; without signal presequence, primers: forward 5'-CACCATATGTTATCTCT CAGAAACGTTCCTG-3', reverse 5'-CACGGATCCTGGAGCGTTTATTTCAGTTAGA-3') and Nf hemerythrin (A0A6A5BXL4; primers: forward 5'-CACCATATGGCCACTAC TATTCCATCACCATTTA-3', reverse 5'-CACGGATCCTTAAAGAACACCCTTG TACTTCATAT-3') were amplified from *N. fowleri* HB-1 cDNA. RNA was isolated using the High Pure RNA Isolation Kit (Roche) and transcribed into cDNA using the High-Capacity cDNA Reverse Transcription Kit (Thermo Fisher Scientific), all according to the manufacturer's protocol. The Nf lysozyme and Nf cystatin genes were subcloned into a pET42b vector (Merck, USA) and expressed with a C-terminal 6×His tag. The Nf hemerythrin gene was subcloned into a pET42b vector with a stop codon, resulting in recombinant protein without any tag. Nf lysozyme and Nf cystatin were expressed in the BL21(DE3) strain of *E. coli* (Merck) in lysogeny broth medium containing 50 μg/mL of kanamycin for 3 hours after induction by 0.5 mM isopropyl-β-D-1-thiogalactopyranoside (IPTG, Sigma-Aldrich) and then purified on Ni-NTA agarose (Qiagen, Germany) under native conditions according to the manufacturer's protocol. Nf hemerythrin was expressed in BL21(DE3) *E. coli* that was grown in M9 minimal medium supplemented with 0.1 mM $FeCl_2$ (Sigma-Aldrich) and 50 μg/mL of kanamycin at 37 °C and 150 rpm. After reaching an optical density of 0.8 at 600 nm, cells were supplemented with additional 0.1 mM $FeCl_2$ and protein production was induced with 0.5 mM IPTG. Cells were further grown overnight at 25 °C. To purify Nf hemerythrin, cells were harvested by centrifugation at 7000 × g for 10 min, resuspended in 20 mM Tris–HCl (Sigma-Aldrich) buffer

pH 7.5 (buffer A) and disrupted by 3 cycles in a French-Press apparatus at 16 000 psi (Thermo Fisher Scientific) in the presence of DNAse (Applichem, Germany). The crude extracts were centrifuged at 25 000 × g for 30 min and at 138 000 × g for 2 h at 4 °C to remove cell debris and membrane fraction, respectively. The soluble extracts were loaded onto a Q-Sepharose Fast-Flow column (70 mL, GE Healthcare, USA), previously equilibrated with buffer A. The proteins were eluted with a linear gradient from buffer A to 20 mM Tris–HCl pH 7.5 containing and 500 mM NaCl (buffer B). The eluted fractions were analyzed by 15% SDS/PAGE and UV–visible spectroscopy and the ones containing the desired protein were pooled and concentrated. The fraction containing NF hemerythrin was then loaded onto a size exclusion S75 column (330 mL, GE Healthcare) equilibrated with 20 mM Tris–HCl, pH 7.5 containing and 150 mM NaCl. The eluted fractions were again analyzed by 15% SDS/PAGE and UV–visible spectroscopy and the ones containing the desired protein were pooled, concentrated, and used in the following assays. The protein and iron content was analyzed in the pure fractions as described before [40].

Polyclonal antibodies against Nf lysozyme were raised as described in [67].

### Size-exclusion chromatography

To determine quaternary structures the purified recombinant proteins were subjected to size-exclusion chromatography. The proteins were loaded on two different separation columns based on their predicted size, Superdex 200 Increase 10/300 GL and Superdex 75 Increase 10/300 GL (both GE Life Sciences, USA). Columns were equilibrated with 20 mM Tris-HCl, 150 mM NaCl (Sigma-Aldrich), pH 7.5. The molecular weight of Nf lysozyme and Nf cystatin eluted from Superdex 200 Increase 10/300 GL column was calculated using calibration curves of the Gel Filtration Standard mixture (Bio-Rad, USA). The molecular weight of Nf hemerythrin eluted from Superdex 75 Increase 10/300 GL column was calculated using calibration curve of Gel Filtration LMW Calibration Kit (Cytiva, USA).

### Lysozyme activity

A simple assay using *M. lysodeikticus* cells as substrate was used to detect the presence of lysozyme activity in recombinant Nf lysozyme. The initial cell suspension of 0.15 mg/ml *M. lysodeikticus* (M3770, Sigma-Aldrich) was prepared in PBS and 2.5 ml of this suspension was pipetted into a cuvette and equilibrated to 25 °C. To the cuvettes containing the cell suspension, 100 µl of PBS was added as a blank, 100 µl of 10 µg/ml (equivalent to 400 units/ml) chicken egg white lysozyme (L6876, Sigma-Aldrich) was added as a positive control, and 100 µl of 10 µg/ml recombinant Nf lysozyme was added as an unknown sample, immediately mixed by inversion, and the change in absorbance at 450 nm was recorded for 5 min.

### Western blot

To analyze the expression of *N. fowleri* lysozyme after co-culture with different types of prey, the cells were continuously cultured (5 passages) with *K. aerogenes*, *M. lysodeikticus*, HT1080 fibrosarcoma cells or axenically. Cells were washed with PBS and protein content was determined using a BCA protein assay kit (Thermo Fisher Scientific). Equal amounts of proteins were separated by SDS-PAGE. Proteins were transferred to nitrocellulose membranes by semidry electroblotting and visualized by Ponceau S staining (0.5% Ponceau S (Merck), 1% acetic acid) to confirm equal sample loading. Primary antibodies at a dilution of 1:100 were used to detect the target proteins. Secondary antibodies were anti-rat IgG conjugated to horseradish peroxidase (Merck) used at 1:5000 dilution and visualized with chemiluminescent peroxidase Substrate-1 (Sigma-Aldrich) on an Amersham Imager 600 (GE Life Sciences) according to the manufacturer's protocol.

## Cystatin inhibitory activity

Recombinant human cathepsin K holoenzyme (a gift from dr. Martin Horn and dr. Michael Mareš, Institute of Organic Chemistry and Biochemistry, Academy of Sciences of the Czech Republic, Prague), cathepsin L holoenzyme from human liver (C6854, Merck), and cathepsin B holoenzyme from bovine spleen (C6286, Merck) were used. The test was performed with 50 nM enzymes and increasing concentrations of Nf cystatin (5 nM – 2 μM) in a reaction buffer composed of 50 mM sodium acetate pH 5.5 + 100 mM NaCl + 0.25 mM dithiothreitol. The total volume of 100 μl was incubated in black flat-bottom microtiter plates (Nunc) for 15 min at RT. Then, 100 μl of 200 μM fluorogenic peptide substrate Z-Phe-Arg-AMC (4003379, Bachem) diluted in 33% dimethylsulfoxide in the reaction buffer was added to the wells. Fluorescence resulting from peptidolytic release of aminomethylcoumarin from the substrate (relative fluorescence units, RFU) was measured for 60 min in 1 min kinetic cycles at 28°C and 355/460 nm excitation/emission wavelengths in Infinite M200 fluorometer (Tecan). Results were expressed as relative peptidolytic activity in the well with Nf cystatin (ΔRFU/min) related to the activity of the enzyme alone taken as 100% (ΔRFU/min, linear part of the curve, no Nf cystatin added). Five measurements were done with cathepsin K, three with cathepsin L, and two with cathepsin B. The $IC_{50}$ of Nf cystatin with each enzyme was calculated and plotted in GraphPad Prism v.10.2.0.

## Spectroscopic methods

The anaerobic reduction of Nf hemerythrin was achieved by sequential additions of sodium dithionite and methyl viologen, followed by buffer exchange on PD-10 desalting column (Cytiva). The protein samples in the fully reduced state and NO-incubated were then analyzed by UV–visible spectroscopy in a Shimadzu UV-1800 spectrophotometer, inside an anaerobic chamber (Coy Lab Products, USA). These samples were then transferred to EPR tubes, frozen in liquid nitrogen, and analyzed by EPR spectroscopy using a Bruker EMX spectrometer equipped with an Oxford Instruments ESR 900 continuous-flow helium cryostat and a perpendicular mode rectangular cavity. The "as purified" samples were prepared aerobically. Protein samples were prepared to final concentrations of 200 μM. EPR spectra were simulated using the program SpinCount [73].

## Amperometric measurement of NO consumption

The NO binding capacity of Nf hemerythrin was determined amperometrically with Clark-type electrode selective for NO (ISO-NOP, World Precision Instruments, USA), under anaerobic conditions in 100 mM Tris–HCl, pH 7.5 containing 5% glycerol and in the presence of an $O_2$ scavenging system (10 mM glucose, 375 nM glucose oxidase and 750 nM catalase) to avoid the formation of other species. Sequential additions of NO (up to 20 μM) were followed by the addition of Nf hemerythrin (5 μM). Stock solutions of 1.91 mM NO were prepared by saturating a degassed 100 mM Tris–HCl, pH 7.5 containing 5% glycerol buffer solution in a rubber seal capped flask with pure NO gas (Air Liquide) at 1 atm on ice: gaseous NO was flushed through a 5 mM NaOH trap to remove higher N-oxides and a second trap with deionized water to remove aerosols. After this, the solution was allowed to equilibrate at room temperature.

## Secretome analysis

*N. fowleri* cells cultured axenically or in long-term co-culture with HT1080 in biological tetraplicates were washed twice with PBS in culture flasks. After the washing, the cells were incubated in glucose medium for 1 hour at 37 °C. After the incubation, the cells were removed from the glucose medium by centrifugation at 1200 × g for 10 min at 4 °C. The resulting

supernatant was centrifuged at 9000 × g for 20 min to remove cell debris, filtered through a 0.22 µm PVDF filter, and centrifuged at 110 000 × g for 90 min to remove microvesicles [74]. Proteins in the final supernatant were precipitated with trichloroacetic acid (TCA, Sigma-Aldrich) in a 1:4 ratio (one volume of TCA to four volumes of supernatant) for 30 min at 4 °C. The precipitated proteins were centrifuged at 40 000 × g for 10 min at 4 °C, washed twice with ice-cold acetone, dried, and stored at −80 °C. Pellets were further analyzed using label-free proteomics as described in the LC-MS method. Data analyses were performed using Perseus software. For each identified protein, the cell localization and secretory pathway signal were predicted using the DeepLoc 2.0 (https://services.healthtech.dtu.dk/services/DeepLoc-2.0/), SecretomeP 2.0 (https://services.healthtech.dtu.dk/services/SecretomeP-2.0/), SignalP 6.0 (https://services.healthtech.dtu.dk/services/SignalP-6.0/) and TargetP 2.0 (https://services.healthtech.dtu.dk/services/TargetP-2.0/). Proteins were sorted into functional groups based on conserved domain predictions from Pfam 36.0 (http://pfam.xfam.org/) and distant homologies detected by HHpred.

The second secretome was prepared from axenically cultured amoebae and long-term co-cultured amoebae directly incubated with HT1080 fibrosarcoma cells in three biological replicates. Prior to the experiment, amoebae were grown to logarithmic phase, washed twice with PBS along with HT1080 cells, and counted on a Guava easyCyte 8HT flow cytometer (Luminex Corporation). Amoebas and HT1080 cells were mixed in a 2:1 ratio (amoeba:fibrosarcoma) in glucose medium and incubated at 37 °C for 1 hour. After incubation, the glucose medium from both conditions was processed in a manner similar to the secretome described previously. Pellets were further analyzed using label-free proteomics as described in the LC-MS method. Data analyses were performed using Perseus software and visualized as a volcano plot using the online tool VolcaNoseR (difference 0.9, significance threshold 2) [75].

## Raman microscopy

The axenically cultured *N. fowleri* cells and those isolated from the mouse brain after 5-days of infection were spun down at 1200 × g, fixed in 1% formaldehyde, and immobilized in 1% low-temperature-melting agarose (Merck) in between a quartz slide and quartz coverslip followed by consecutive sealing with CoverGrip (Biotium, USA) [76]. We measured 4 cells per sample with consistently reproducible results per phenotype (S11 Fig).

Raman mapping was performed using Witec alpha 300 RSA system equipped with a 40 mW 532 nm laser, UHTS300S detector using 600 g/mm dispersion grating, and water immersion objective 60× UPlanSApo, NA 1.2 (Olympus, Japan). Spatial resolution for imaging was set to 200 nm. The integration time per pixel varied between 0.07 and 0.2 s for spectral mapping [77]. To improve the signal-to-noise ratio, we acquired single spectra to better characterize the spectral profile of diverse cell compartments using an integration time of 0.5 s over up to 200 accumulations [78].

Data processing was done in WITec Project 6.0 software (WITec, Germany) using cosmic ray removal, spectra cropping, polynomial and shape background subtraction, and the spectral decomposition tool as previously described in [79]. The single spectra have been exported, normalized for maximum, and plotted by SigmaPlot 12.5.

## Supporting information

**S1 Video. Live holotomographic imaging of unlabeled *Naegleria fowleri* and HT1080 fibrosarcoma cells co-culture over a period of 6 hours showing cytopathogenicity of the amoebae.**
(AVI)

**S1 Fig. Live imaging of CFSE-labeled *Naegleria fowleri* (green) and HT1080 fibrosarcoma cells with tdTomato protein in cytosol (magenta) co-cultures showing the process of *Naegleria* adhesion to human cells.** Ingested parts of HT1080 cells are visible in *Naegleria* vacuoles. Nuclei were labeled with Hoechst 33342 (blue). Scale bar=10 μm.
(PDF)

**S2 Fig. Representative flow cytograms of CFSE-labeled *Naegleria fowleri* (green), HT1080 cells expressing tdTomato (red) and their co-culture after pre-incubation with different inhibitors (CK-666, wortmannin, piceatannol and PP2).** (A) Flow cytograms of CFSE-labeled *N. fowleri* incubated with selected concentrations of inhibitors for 3 hours, showing no changes in the gated amoeba population. (B) Flow cytograms of HT1080 tdTomato cells incubated with selected concentrations of inhibitors for 3 hours showing no changes in the gated mammalian cell population. (C) Flow cytograms of CFSE-labeled *N. fowleri* preincubated with inhibitors in co-culture with HT1080 tdTomato cells. Cytopathogenicity is indicated by the number of *Naegleria* with ingested cell parts, represented by the red fluorescence of the tdTomato (orange) after 3 hours of co-culture. Preincubation with DMSO was used as a control.
(PDF)

**S3 Fig. Protein sequence and structure of *Naegleria fowleri* lysozyme (A0A6A5BDE0) predicted by Alphafold.** Lysozyme domain is shown in purple; peptidoglycan binding domain is shown in yellow, and disordered regions are shown in gray.
(PDF)

**S4 Fig. Proteins upregulated in *Naegleria fowleri* isolated from mice brains in comparative proteomics sorted into functional categories using HHpred annotation.**
(PDF)

**S5 Fig. Protein sequence and structure of *Naegleria fowleri* cystatin (A0A6A5BXL1) predicted by Alphafold.** Cystatin domain is shown in purple with cystatin motif in blue and signal presequence is shown in yellow.
(PDF)

**S6 Fig. Chromatograms from size-exclusion chromatography of recombinant *Naegleria fowleri* cystatin, lysozyme and hemerythrin.** The molecular weights calculated for each significant peak in the chromatograms are shown within tables. Molecular weights were calculated from calibration curves of the Gel Filtration Standards (Bio-Rad, USA) and for hemerythrin from Gel Filtration LMW Calibration Kit (Cytiva, USA). The peak of the standards along with their molecular weight is listed at the top of the chromatographs. The inset in each chromatograph shows SDS-PAGE of the recombinant protein.
(PDF)

**S7 Fig. Enzymatic activity of lysozyme and control of protein loading.** (A) Enzymatic activity of recombinant *Naegleria fowleri* lysozyme (blue line) and chicken egg white lysozyme (red line). A decrease in absorbance at 450 nm over time is associated with lysozyme enzymatic activity. The black line represents a blank sample without enzyme addition. (B) Control of protein loading for the immunoblot of *N. fowleri* lysozyme in Fig 4, stained by Ponceau S. ax.: total lysate from axenically cultured *N. fowleri*, g-: *N. fowleri* co-cultured with *Klebsiella aerogenes*, g+: *N. fowleri* co-cultured with *Micrococcus lysodeikticus*, fib.: *N. fowleri* co-cultured with HT1080 fibrosarcoma cells.
(PDF)

**S8 Fig. Structural alignment of the *Naegleria fowleri* hemerythrin (A0A6A5BXL4, yellow) predicted by Alphafold and the crystallographic structure of the *Methylococcus capsulatus***

**hemerythrin (PDB 4XPX, blue).** Protein alignment of the hemerythrins with the iron-binding amino acids highlighted.
(PDF)

**S9 Fig. Representative flow cytograms of CFSE labelled (A) axenic *Naegleria fowleri* and (B) long-term co-cultured *N. fowleri* (green) in co-culture with HT1080 cells expressing tdTomato (red).** Cytopathogenicity is indicated by the number of *Naegleria* with ingested cell parts, represented by the red fluorescence of the tdTomato (orange) after 1 and 3 hours of co-culture.
(PDF)

**S10 Fig. Long-term co-culture of *Naegleria fowleri* with mammalian cells affects amoeba cytopathogenicity but has no effect on its proliferation rate and trogocytosis.** (A) Graph of doubling time of axenically cultured *N. fowleri* (axenic) and long-term co-cultured *N. fowleri* (co-cultured) in host cell-free medium. (B) Graphs showing changes in the percentage of CellTrace Far Red-labeled axenic or long-term co-cultured *N. fowleri* (Nf CelltraceFR) with ingested either cytosolic eGFP or membrane-targeted eGFP-CAAX from HT1080 cells after 3 hours of co-incubation as measured by flow cytometry. (** p-value<0.01) (C) Live imaging of CellTrace Far Red-labeled *N. fowleri* (magenta) and HT1080 fibrosarcoma cell with eGFP-CAAX (green) co-culture showing the ingestion of human cell plasma membrane by amoebae. Scale bar=10 μm.
(PDF)

**S11 Fig. Raman chemical maps of *Naegleria fowleri* cultivated axenically (A) and isolated from a mouse brain (B).** These maps illustrate significantly different phenotypes of both cultivation conditions based on the major spectral components representing the cytoplasm and nucleus detectable in most of the measured cells (with some exceptions when the confocally limited optical section missed the nucleus) or ingested cells and lipid droplets detected in *Naegleria* cells isolated from the brain.
(PDF)

**S1 Table. Comparative proteomic analysis of axenically cultured *Naegleria fowleri* and *Naegleria* co-cultures with bacteria (*Klebsiella aerogenes*), short-term (6 hours), long-term conditions (5 passages) with human fibrosarcoma cells HT1080 and amoebae isolated from the brains of infected mice.**
(XLSX)

**S2 Table. List of upregulated proteins from the comparative proteomic analysis of *Naegleria fowleri* isolated from mice brains, annotated using HHpred.**
(XLSX)

**S3 Table. Summary of the top (by iPtm) models from AlphaFold-multimer analysis of hypothetical interactions between Nf cystatin and cysteine proteases from *Naegleria fowleri*.**
(XLSX)

**S4 Table. Summary of the top (by iPtm) models from AlphaFold-multimer analysis of hypothetical interactions between Nf cystatin and cysteine proteases from human.**
(XLSX)

**S5 Table. Secretome of axenically maintained *Naegleria fowleri* and long-term co-cultured *N. fowleri*.**
(XLSX)

**S6 Table. Secretome of axenically maintained *Naegleria fowleri* and long-term co-cultured *N. fowleri* after 1 h incubation with human cell line HT1080.**
(XLSX)

**S7 Table. Description of proteomic data analysis.**
(DOCX)

## Acknowledgments

Thanks to Pavel Talacko and Karel Harant from the Laboratory of Mass Spectrometry, Biocev, Charles University, Faculty of Science, where the proteomic and mass spectrometric analysis was performed. Thanks to Peter Mojzeš from the Institute of Physics, Faculty of Mathematics and Physics, Charles University, where the Raman imaging was performed. Thanks to Ivan Hrdý from Department of Parasitology, Faculty of Science, Charles University, BIOCEV, for the help with FPLC. The authors acknowledge Imaging Methods Core Facility at BIOCEV, institution supported by the MEYS CR (LM2023050 Czech-BioImaging) for their support & assistance in this work.

## Author contributions

**Conceptualization:** Ronald Malych, Robert Sutak.

**Data curation:** Filipe Folgosa, Jana Pilátová, Vít Dohnálek, Jan Mach, Magdaléna Matějková, Robert Sutak.

**Formal analysis:** Ronald Malych, Filipe Folgosa, Vít Dohnálek, Magdaléna Matějková.

**Funding acquisition:** Robert Sutak.

**Investigation:** Ronald Malych, Jana Pilátová, Libor Mikeš, Jan Mach.

**Methodology:** Filipe Folgosa, Libor Mikeš, Vít Dohnálek.

**Supervision:** Robert Sutak.

**Validation:** Vladimír Kopecký, Pavel Doležal.

**Visualization:** Ronald Malych, Filipe Folgosa, Jana Pilátová.

**Writing – original draft:** Ronald Malych, Robert Sutak.

**Writing – review & editing:** Ronald Malych, Filipe Folgosa, Jana Pilátová, Libor Mikeš, Vladimír Kopecký, Pavel Doležal, Robert Sutak.

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
