## [Decision Letter · Decision Letter 0]

12 Nov 2024

PPATHOGENS-D-24-02198Eating the brain - a multidisciplinary study provides new insights into the mechanisms underlying the cytopathogenicity of Naegleria fowleri.PLOS Pathogens Dear Dr. Sutak, Thank you for submitting your manuscript to PLOS Pathogens. After careful consideration, we feel that it has merit but does not fully meet PLOS Pathogens's publication criteria as it currently stands. Therefore, we invite you to submit a revised version of the manuscript that addresses the points raised during the review process. Please submit your revised manuscript within 60 days Jan 11 2025 11:59PM. If you will need more time than this to complete your revisions, please reply to this message or contact the journal office at plospathogens@plos.org.  Please include the following items when submitting your revised manuscript:* A rebuttal letter that responds to each point raised by the editor and reviewer(s). You should upload this letter as a separate file labeled 'Response to Reviewers '. This file does not need to include responses to any formatting updates and technical items listed in the 'Journal Requirements' section below.* A marked-up copy of your manuscript that highlights changes made to the original version. You should upload this as a separate file labeled 'Revised Manuscript with Track Changes '.* An unmarked version of your revised paper without tracked changes. You should upload this as a separate file labeled 'Manuscript '. If you would like to make changes to your financial disclosure, competing interests statement, or data availability statement, please make these updates within the submission form at the time of resubmission. Guidelines for resubmitting your figure files are available below the reviewer comments at the end of this letter. We look forward to receiving your revised manuscript. Kind regards, Tracey J. LambSection EditorPLOS Pathogens Tracey LambSection EditorPLOS Pathogens Michael Malim

Editor-in-Chief

PLOS Pathogens

orcid.org/0000-0002-7699-2064 **Additional Editor Comments (if provided):**     **Journal Requirements:****Reviewers' Comments:**  Reviewer's Responses to Questions

**Part I - Summary**

Reviewer #1: The manuscript by Malych et al. investigates how Naegleria fowleri amoebae interact with mammalian host cells. This is an understudied topic, and more work in the area is needed to understand this deadly pathogen and ultimately devise new therapeutics. The authors first show that actin is important for consumption of human cells using flow cytometry. They then compare proteomes of amoebae cultured in different conditions, and identify and characterize proteins that are upregulated in mouse-passaged amoebae. They go on to analyze secreted proteins, and end with Raman Microspectroscopy comparing axenic to mouse-passaged amoebae. The variety of techniques used is quite impressive, the flow of the paper is logical, and the findings could influence the manner in which Naegleria’s cell biology is studied (in co-culture vs mouse, see point 1). This work would be of broad interest to researchers studying eukaryotic pathogens, or comparing cell-based and mouse models. This work would also be very useful to the Naegleria community. While we appreciate the amount of work that went into this study, we have a few issues that we believe should be addressed.

Reviewer #2: The goal of this study by Malych et al. is to identify factors at the protein level that uniquely empower virulence within the Naegleria fowleri compared with other Naegleria species. The authors use an in vitro co-culture system with human fibrosarcoma feeder cells and N. fowleri of different passage histories (axenic, mouse, etc) to attempt to define proteins that could be involved in contact-dependent and independent cytopathogenicity. Several complementary in vitro, in silico, and proteomic approaches were used, with some outcomes being more conclusive than others. In the absence of in vivo tests, the relevance of these findings are ultimately difficult to know, but they nonetheless add to our overall understanding of Naegleria biology. There are several areas where the authors could/should improve this manuscript.

Reviewer #3: This manuscript effectively explores various potential virulence factors that the amoeba can utilize under different conditions.

I have few important questions to address. Furthermore, I want to highlight that the introduction offers a comprehensive and ordered overview of the history of some important virulence factors previously described in the amoeba.

1. Is there a specific reason you chose to use this fibrosarcoma cell line?

2. you mention in lines 85 to 88 that "we investigated the mechanism of contact-dependent cytopathogenicity using inhibitors of actin and several protein kinases related to cellular processes such as motility and adhesion". The question is:

Why were only actin and kinase inhibitors chosen to assess the contact-dependent mechanism? What led to the decision to consider solely actin as contact-dependent? Do you consider that some of analyzed enzymes play a role as contact-dependent mechanisms?

3. In lines 139 you mention that " We further focused on the analysis of the 18 upregulated proteins that were shared between the long-term fibrosarcoma co-culture and brain-isolated amoebae as these are likely to be involved in the

interaction with the host"

What reasons could explain why proteins previously reported as contact-dependent for the amoeba, such as integrin-like proteins, membrane protein (Mp2CL5), or Nfa1, do not appear on this particular list (18 upregulated proteins)?

4. In line 152 you mention that "This lysozyme (here after named as Nf lysozyme) contains two domains: a lysozyme and a peptidoglycan binding domain (protein sequence and structure prediction is depicted in S3 Fig).

5. In line 172 tha athors mention that "the expression of Nfa1 did not significantly change between conditions, but the paralog of this protein, hemerythrin (A0A6A5BXL4), was identified only in brain-isolated N. fowleri"

Why did the expression of Nfa1 remain largely unchanged despite the direct interaction with the brain? The evidence has been supported that this protein serves as a significant adhesion factor in vitro. What implications does this have for this result?

How would you explain and through which domain of the enzyme (lysozyme) the union with some host protein would be carried out (contact-dependent manner), and if you could only suggest some host receptor according to this type of enzyme?

6. In line 189-192 the authors report Nf cystatins structures. Do you think that these enzymatic structures could be participating in contact-dependent binding to a receptor on the host cells?

7. In lines 203-205 the authors mention that " Immunoblots showed an increase in Nf lysozyme protein levels after long-term co-culture with both gram-positive bacteria and fibrosarcoma cells, suggesting its involvement in the interaction with the prey".

It would likely be more appropriate to state that the protein showed an increase in antibody recognition intensity, suggesting a more active involvement of this protein.

When the authors refer to "prey," are they talking about bacteria specifically, or do they mean all the cells they interacted with, including mammalian cells? Additionally, which band corresponds to the protein with an approximate molecular weight of 25 kDa (fig 4 A), and what proteins does it represent?

I am unsure if the image quality and resolution are the final version, but they are not the best.

**Part II – Major Issues: Key Experiments Required for Acceptance**

Reviewer #1: 1. The finding that the proteome of mouse passaged amoebae is so different from long-term co-cultured amoebae seems potentially very impactful. Does this throw into question the usefulness of experiments using co-culture systems? We think this could be a major take-away from this work that is worth discussing. While this major point does not involve experiments, I think addressing this point would strengthen the impact of the manuscript.

2. I have some hesitations about the flow cytometry data. For figures 1A, 1C, and 5D, how were the gates for red or green fluorescence determined? It would be appropriate to run amoebae alone (+/- inhibitors) and human cells alone (subject to the same treatments and dyes but without amoebae) to determine the borders of the gates. The imaging shown in Movie S1 is beautiful, but if I’m not mistaken, there is detectable CFSE background in the human cells, which could result in human cells that have taken up some amount of residual CFSE being counted as amoeba that have consumed mammalian cells. Further, the inhibitors used could lead to changes in either the amoebae or mammalian cells that may impact the fluorescence (e.g. if it results in any cell death or leakage). I think it is worth being a bit cautious with this gating, because if I understand the methods correctly, the amoebae and drugs were mixed 1:1 with human cells, which would mean the human cells are also subjected to a 3 hour incubation with these inhibitors, which could change their biology, especially at some of the concentrations used (also see minor point 4).

Reviewer #2: The authors show that inhibition of actin networks impairs feeding behavior, as has been reported in N. gruberi. The authors should explain how know that actin is used by Naegleria in general for feeding advances our understanding of why N. fowleri is apparently the only species of Naegleria that is pathogenic.

For experiments using mouse passaged amoeba, the authors should state how long after mouse infection amoeba harvested and how long after harvest from mouse are amoeba used.

In Figure 1C—are inhibitors washed away from amoeba or are they maintained in inhibitor solution while plated with feeders cells ? Methods don’t specifically address this.

Can the authors functionally demonstrate differences in production of protease/proteases from using different N. fowleri conditions with feeder cells/axenic etc conditions?

Naegleria have not evolved to feed on mammalian tissues while facing an mammalian immune system, they are accidental pathogens. With that in mind, it would be useful to frame the production of these factors less through the lens of attacking the host, but rather upregulating feeding mechanisms that are appropriate for the food source. Blocking cysteine protease activity and scavenging free radicals would conceivably allow N. fowleri to live longer in a host, but what “normal” purpose might these serve. Why would amoeba be using these in their natural environment compared with other species of N. fowleri?

The results presented here and in prior literature indicate that some N. fowleri proteins have the ability to inhibit some proteases from different mammalian hosts. How do the authors interpret lines 206-209 (i.e what does this really mean?) There is discussion of cystatins, but not how the authors envision these highly variably pairings of activity/function could result in a functional impacts in vivo---especially when there are many proteases that would likely not be inhibited?

Figure 5: The attempt to disentangle cell destruction via trogocytosis vs that done by cell lysis is an admirable and much needed line of investigation. It’s not clear how the work in Figure 5 answers this.

On line 290, it’s unclear what “…and the percentage of HT1080 cells after 1 and 3 hours of co-culture….” Means. Are words missing?

Can this assay (in Figure5) differentially report cells consumed via trogocytosis or lysis? If so, this should be clearly stated. If not, could something similar be done with cells that express either membrane or organelle argeted fluorescent proteins? Or perhaps with physical separation of amoeba and feeders so that trogocytosis is impossible.

Along similar lines, do amoeba need to contact feeders in order to upregulate these proteins? Or can they sense their environment and produce cytopathic factors after “smelling” cells nearby.

When comparing axenic and mouse passaged (or long term feeder passage), like in Fig5, is the actual proliferation rate different? Or total number of amoeba cells produced when all feeders cells killed/consumed different? It would be useful to understand what functional advantages these more “virulent” cells have, as the differences between groups at 1 and 3hrs are quite modest. Are passaged cells more efficient at killing or more efficient at “growing”?

While I’m admittedly ignorant of Raman microscopy, how can one know that nucleic acids are from mitochondria and not something else?

Reviewer #3: No Major issue

**Part III – Minor Issues: Editorial and Data Presentation Modifications**

Reviewer #1: Minor points:

1. For flow cytometry data, was any initial gating done to remove debris and doublets?

2. Was microscopy done on CK-666 treated cells or PP2-treated cells? If this data is available, it would be nice to see in the publication.

3. I do not think this is necessary for publication, but an experiment the authors may wish to consider to strengthen their claims and distinguish trogocytosis-induced damage from damage done by secreted proteases could be to treat mammalian cells with supernatants from axenic or co-cultured samples and assess damage of mammalian cells (perhaps with AnnexinV staining, or some other dye). This is just a suggestion, and I understand if it is beyond the scope!

4. Is there any evidence that wortmannin, piceatannol, or PP2 are effective in Naegleria? The amounts used seem quite high relative to the concentrations typically used in mammalian cells (eg. PP2 is often used at <5 nM, but is used here at 10 uM).

5. After a 3 hour incubation with CK666, which may also prevent macropinocytosis in axenic cells, did the cells differentiate into flagellates? If so, this may have resulted in less trogocytosis. A sentence in the text to address if this may have occurred would be sufficient.

6. The discussion could be shortened; some parts feel repetitive with the results.

7. The results section title “Long-term co-culture of N. fowleri with mammalian cells increases protease secretion and trogocytosis” seems misleading. There were more proteases secreted by axenic cells (18) than co-cultured cells (14) in Figure 5A. Did I miss something?

Minor errors/typos the authors may wish to correct:

Line 94 add [a] before significant role

Line 314 typo “cytoplam”

Line 326 “dis the amoeba”

Line 167 italics for H. sapiens

Lines 28 says “amoeba cell ingestion” while lines 95-96 say “host cell ingestion.” It may be worth using consistent language or picking a clearer term (like ingestion of host cells by amoebae)

Line 105: replace “on” with “the”

Line 220-221, is this intended to be “UV light”?

Reviewer #2: Line 314, cytoplasm is misspelled

On line 326, the word “dis” is written and presumably there is something else missing.

Figures 5D and 5C appear to be swapped (with regard to in text reference)

Reviewer #3: (No Response)

PLOS authors have the option to publish the peer review history of their article (what does this mean? ). If published, this will include your full peer review and any attached files.

**Do you want your identity to be public for this peer review?** For information about this choice, including consent withdrawal, please see our Privacy Policy .

Reviewer #1: No

Reviewer #2: No

Reviewer #3: No

---

## [Decision Letter · Decision Letter 1]

19 Feb 2025

Dear Dr. Sutak,

We are pleased to inform you that your manuscript 'Eating the brain - a multidisciplinary study provides new insights into the mechanisms underlying the cytopathogenicity of Naegleria fowleri.' has been provisionally accepted for publication in PLOS Pathogens.

Best regards,

Tracey J. Lamb

Section Editor

PLOS Pathogens

Tracey Lamb

Section Editor

PLOS Pathogens

Sumita Bhaduri-McIntosh

Editor-in-Chief

PLOS Pathogens

orcid.org/0000-0003-2946-9497

Michael Malim

Editor-in-Chief

PLOS Pathogens

orcid.org/0000-0002-7699-2064

Reviewer Comments (if any, and for reference):

Reviewer's Responses to Questions

**Part I - Summary**

Reviewer #1: All of our concerns have been addressed. We applaud the authors on an excellent manuscript!

Reviewer #2: (No Response)

**Part II – Major Issues: Key Experiments Required for Acceptance**

Reviewer #1: (No Response)

Reviewer #2: (No Response)

**Part III – Minor Issues: Editorial and Data Presentation Modifications**

Reviewer #1: (No Response)

Reviewer #2: (No Response)

PLOS authors have the option to publish the peer review history of their article (what does this mean? ). If published, this will include your full peer review and any attached files.

**Do you want your identity to be public for this peer review?** For information about this choice, including consent withdrawal, please see our Privacy Policy .

Reviewer #1: No

Reviewer #2: No

---

## [Editor Report · Acceptance letter]

Dear Dr. Sutak,

We are delighted to inform you that your manuscript, "Eating the brain - a multidisciplinary study provides new insights into the mechanisms underlying the cytopathogenicity of Naegleria fowleri.," has been formally accepted for publication in PLOS Pathogens.

Best regards,

Sumita Bhaduri-McIntosh

Editor-in-Chief

PLOS Pathogens

orcid.org/0000-0003-2946-9497

Michael Malim

Editor-in-Chief

PLOS Pathogens

orcid.org/0000-0002-7699-2064